# A Rho GTPase-effector ensemble governs cell migration behavior

Heeyoung Lee[1,11], Sangkyu Lee[2,11], Yeji Seo[3,11], Dongsan Kim[4,11], Yohan Oh [5,6], Juae Jin [1], Bobae Hyeon [1], Younghyun Han [4], Hyunjun Kim[7], Yong Jin Lee[3], Ho Min Kim [1], Gabsang Lee [8] ✉, Kwang-Hyun Cho [4] ✉ & Won Do Heo [1,9,10] ✉

How can a cell navigate its environment without any external cues? Since such cues are not always present in the environment, cells rely on internal machinery to explore their surroundings. Although Rho GTPases are known for orchestrating cell motility, the intrinsic Rho GTPase-effector mechanisms governing spontaneous migration remain incompletely understood. Here we show an imaging-based method that profiles protein-protein interactions (PPIs) through phase-separated condensates. By applying this method to hundreds of interaction profiles between Rho small GTPases and their effector proteins, we uncovered two intrinsic mechanisms governing cell migration. Formin-like protein (FMNL) determines the front of the cell by restricting Cdc42 activity, establishing front-rear polarity. In contrast, Rac1-ROCK-interaction-mediated arc stress fiber formation at the front inherently enables spontaneous directional changes and enhances cellular responses to external cues. Our findings elucidate the intricate roles of the Rho GTPase-effector ensemble that governs cell migration behavior, revealing an intrinsic program for efficient motility strategies.

Cell motility is crucial to various physiological and pathological contexts, including embryonic development, immune responses, wound healing, and cancer metastasis[1]. The molecular mechanisms governing directional cell migration in response to external stimuli (e.g., chemical or mechanical gradients) are well established[2–4]. In response to external cues, cells establish front-rear polarity through asymmetric re-distribution of key components, especially Rho GTPase-regulated actin network rearrangement[5–7]. At the front of the cell, Rac and Cdc42 control branched and elongated actin filaments leading to the production of lamellipodia (LP) or filopodia (FP), respectively[8,9]. They also control the assembly of adhesions at the leading edge[10–12]. In contrast, at the rear of the cell, RhoA induces membrane retraction and the disassembly of adhesions through ROCK-mediated actomyosin contraction[13,14].

While the roles of Rho GTPases in directed migration in response to external stimuli are well characterized, their functions in spontaneous migration—where cells must establish, maintain, or break polarity without external cues—remain unclear. In this context, key

[1]Department of Biological Sciences, Korea Advanced Institute of Science and Technology (KAIST), Daejeon, Republic of Korea. [2]Center for Cognition and Sociality, Institute for Basic Science, Daejeon, Republic of Korea. [3]Department of R&D, Hulux, Seongnam, Republic of Korea. [4]Department of Bio and Brain Engineering, Korea Advanced Institute of Science and Technology (KAIST), Daejeon, Republic of Korea. [5]Department of Biomedical Science, Graduate School of Biomedical Science & Engineering, Hanyang University, Seoul, Republic of Korea. [6]Department of Biochemistry and Molecular Biology, College of Medicine, Hanyang University, Seoul, Republic of Korea. [7]Department of Biology, Johns Hopkins University, Baltimore, MD, USA. [8]Institute for Cell Engineering, Johns Hopkins University School of Medicine, Baltimore, MD, USA. [9]Department of Cognitive Brain Science, Korea Advanced Institute of Science and Technology (KAIST), Daejeon, Republic of Korea. [10]KAIST Institute for the BioInnovation (KIBI), Korea Advanced Institute of Science and Technology (KAIST), Daejeon, Republic of Korea. [11]These authors contributed equally: Heeyoung Lee, Sangkyu Lee, Yeji Seo, Dongsan Kim. ✉e-mail: glee48@jhmi.edu; ckh@kaist.ac.kr; wondo@kaist.ac.kr

questions include which Rho GTPases determine whether a cell maintains persistent movement or initiates turning, and how their effector interactions regulate these decisions. Understanding these intrinsic regulatory mechanisms is essential to elucidate how cells adaptively control migration behavior.

To address these questions, we developed INSPECT (INtracellular Separation of Protein Engineered Condensation Technique), an imaging-based method that detects protein-protein interactions via phase-separated synthetic condensates formed by multi-subunit proteins. Using INSPECT, we screened the direct interactions of Rho GTPases and their effectors and revealed that Rho GTPase-effector ensembles play central roles in migration behavior. Interestingly, Rac1 and Cdc42, known to work mainly at the front, exhibited similar interaction profiles but differed only in their interactions with ROCK and FMNL.

Specifically, we found that the interaction between FMNLs and Cdc42 restricts the molecular activity of Cdc42, reinforcing and maintaining front-rear polarity. This is then followed by a Rac/ROCK-mediated frontal contractility that induces arc stress fiber (SF) formation and triggers orthogonal turning, enabling spontaneous directional changes. Moreover, we observed that Rac/ROCK-mediated contractility is essential for environmental sensing and responding by influencing Rac1 immobilization at the adhesion, facilitating YAP nuclear translocation, and regulating migration speed. Our data reveal that intrinsic machinery plays a central role in establishing front-rear polarity (Cdc42-FMNL) and spontaneous directional changes (Rac-ROCK). We also demonstrate the dual role of Rac-ROCK in cell motility, inherently providing a self-regulating system with both positive and negative regulation that prevents directional movement. Our findings suggest that cell motility is not a simple matter of moving in a straight line or randomly but rather a sophisticated balance between persistent or directional changes controlled by intrinsic machinery comprised of an ensemble of Rho GTPase and effector proteins.

## Results

### Development of INSPECT to identify protein-protein interactions using phase-separated condensates

To detect and visualize direct PPIs using phase-separated condensates, we used genetically engineered ferritin (FT)[15] and red fluorescent protein (RFP) to individually conjugate prey and bait to different MPs (Supplementary Fig. 1a). We designed constructs encoding FRB-fused FT (FRB-CFP-FT) and a series of FKBP-fused RFPs of oligomeric states ranging from monomer to tetramer and measured the proportion of condensate-forming cells in the presence of rapamycin. We observed condensate formation for tetramers involving LanRFP and DsRed. We selected DsRed as the partner for FT because it yielded the most efficient condensate formation (Fig. 1a, b). Rapamycin treatment triggered the cytosolic dispersion of FKBP-DsRed and FRB-CFP-FT, leading to condensation within a few minutes ($t_{1/2}$ = 2.4 min at 500 nM rapamycin) (Fig. 1c and Supplementary Fig. 1b). Condensate formation was concentration-dependent, with the half-maximal time ($t_{1/2}$) decreasing from $417.2 \pm 44.6$ s at 50 nM Rap to $219.9 \pm 25.5$ s and $146.9 \pm 19.9$ s at 100 nM and 500 nM Rap, respectively (Supplementary Fig. 1c, d). The condensate numbers remained consistent regardless of the rapamycin concentration but showed strong correlation with the fluorescent signal, indicating that they are determined by the protein expression level (Supplementary Fig. 1e). The formation of synthetic condensates via the FKBP-FRB interaction was specifically blocked by a competitor (FK506) (Supplementary Fig. 1f). Additionally, fluorescence recovery after photobleaching (FRAP) analysis revealed that ~99% of the proteins were immobile (Supplementary Fig. 1g, h). This result indicates that most molecules within the condensates were tightly interconnected and restricted in movement.

Applying INSPECT to PPIs, we next tested interactions between the Ras-binding domain (RBD) of Raf1 and HRas in both

its active (Q61L) and inactive (S17N) states. Condensates were formed with only active HRas, demonstrating the INSPECT system's high specificity (Fig. 1d). We then examined interactions between Raf1 and 28 Ras family GTPases. Using INSPECT, we were able to identify the 13 previously known positive pairs with no false positives/negatives, emphasizing the system's reliability for detecting PPIs in cells (Supplementary Fig. 1i)[16,17]. To quantify PPIs using INSPECT, we introduced a comprehensive metric called the Protein Interaction Index (PI) (Fig. 1e), which we calculated based on condensate formation efficiency and signal-to-noise ratio (SNR). Using the PI, we assessed interactions between Raf and Ras family GTPases. Consistent with previous findings, Rap1A and Rap1B showed lower PI values than other Ras GTPases (Fig. 1f). This suggests INSPECT PI values can be used to compare the relative strength or affinities of various protein interactions on a normalized scale.

INSPECT also visualizes ternary protein complexes by co-expressing fluorescent protein conjugated intermediated (I) proteins with Prey-CFP-FT and DsRed-Bait (Supplementary Fig. 2a). We monitored condensate formation using the well-characterized allosteric interactions of Ras (GDP- and GTP-bound) and the Ras-specific guanine nucleotide exchange factor (GEF) Sos[18,19]. We observed condensate formation with the connection of Sos to MP-fused active and inactive HRas (Supplementary Fig. 2b). These condensates persisted even after FT and DsRed pair exchange (Supplementary Fig. 2c). However, when we tested active RRas, which does not participate in Sos allosteric interaction, instead of active HRas, no condensates formed (Supplementary Fig. 2d).

### Screening of interactions between Rho GTPases and their effector proteins

Next, we used INSPECT to compare the interactions between active forms of Rho GTPases and their downstream effector proteins (Supplementary Table. 1). Rho GTPases regulate multiple cellular processes by binding to different effector proteins[20–22], but their activation of multiple effectors, some of which exhibit redundancy, makes it difficult to understand the unique functions of each GTPase. Only a systematic analysis and quantitative comparison of the interactions between Rho GTPase and their various effectors can clarify this issue. Thus, using INSPECT, we investigated the interactions among 285 pairs formed by constitutively active 15 Rho GTPases and 19 cell migration-related effector proteins.

To observe Rho GTPases in the cytosol, we removed the CAAX motifs (Supplementary Fig. 3a), but some, especially those with polybasic tail regions, still showed nuclear localization. In particular, RhoD showed self-condensation even in the absence of effector proteins, possibly due to an ability to homo-oligomerize. We conjugated Rho GTPases to DsRed instead of FT to inhibit further amplification through self-condensation. Once activated, Rho GTPases bind various downstream effector proteins through the G protein-binding domain (GBD). Thus, we conjugated the GBD regions of various effector proteins to FT (Supplementary Fig. 3b). We successfully observed co-localized condensate formation even for Rho GTPases with unique localization or self-condensation features, indicating that these features did not prevent their interaction with effector proteins (Supplementary Fig. 4a). In this analysis, we identified 139 positive interactions, including 19 additional pairs detected using INSPECT (Supplementary Fig. 4a, b).

We noted several things while considering the clustering in the interaction map (Fig. 1g). First, Cdc42 generated condensates with all the tested effector proteins, while Rac proteins generated condensates with most of the same downstream effectors except FMNL2/3. Second, although ROCKs can bind to Rho, Cdc42, and Rac proteins, the observed PI values indicate ROCKs bind more tightly to RhoA or Rac1 than Cdc42 (Fig. 1h, i).

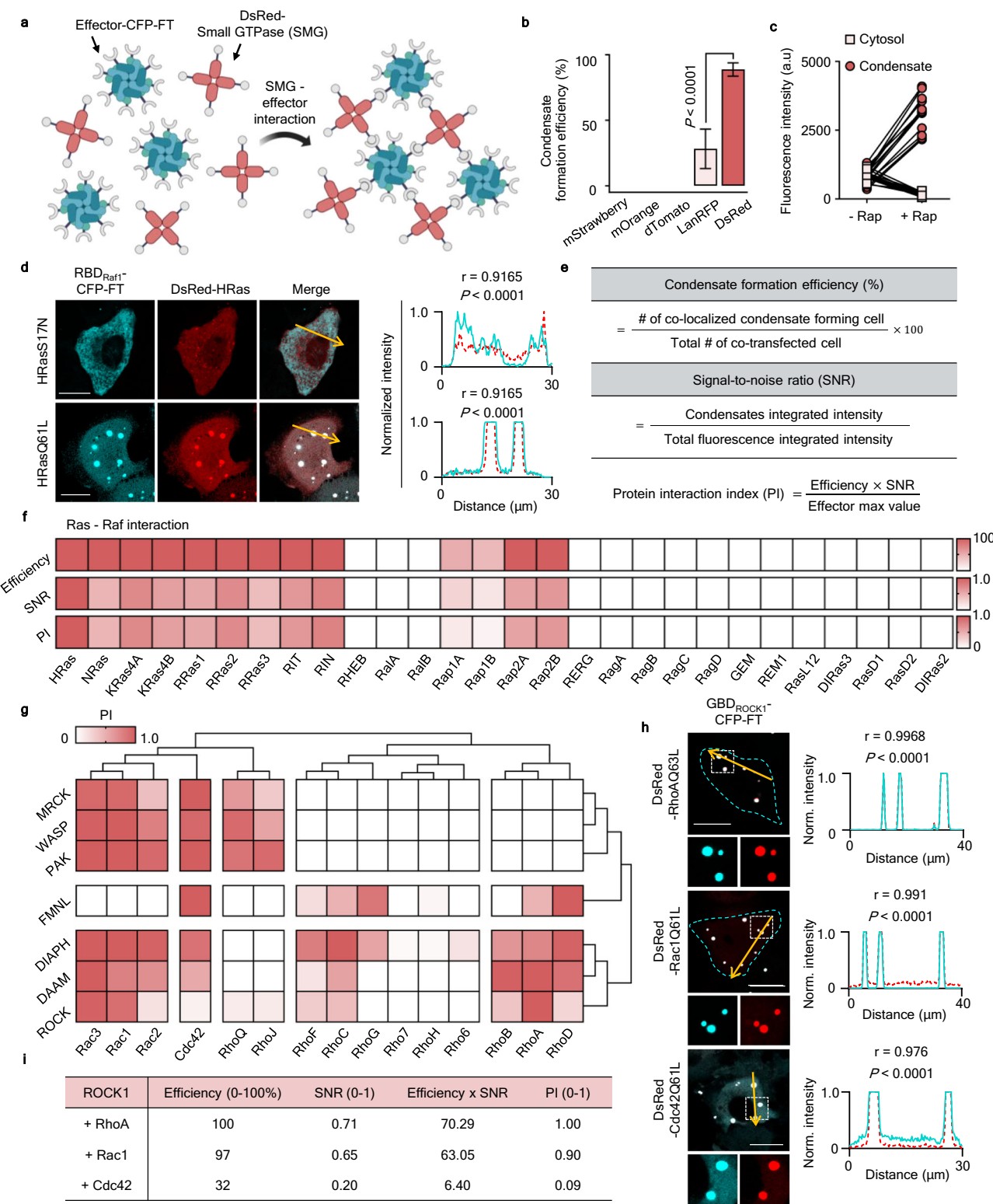

The simplified and summarized interaction profile not only categorizes RhoA apart from Rac1 and Cdc42 but is also consistent with their functional differences (Fig. 2a). In particular, Rac1 and Cdc42 share overlapping interactions but also exhibit distinct binding specificities. We selected three representative effectors (Arp2/3, FMNLs, ROCKs) based on their interactions and functions that partially overlapped with any two proteins among RhoA, Rac1, and Cdc42 but that also showed no or weak interactions with the third remaining protein and defined this set as an ensemble (Supplementary Fig. 4c).

We hypothesized that such an ensemble of Rho GTPases and their effectors determine the specific functions regulated by the individual GTPases. To directly test the hypothesis, we generated ROCK1/2- or FMNL2/3-deleted fibroblasts differentiated from human embryonic stem cells (hESCs) (Supplementary Fig. 5a–d). Then, we tested whether the ROCK or FMNL determined the unique properties of Rac and Cdc42. ROCK1/2-deleted cells did not form LP even when expressing constitutively active Rac1 (Supplementary Fig. 5e–g). In contrast, FMNL2/3-deleted cells expressing constitutively active Cdc42 showed

**Fig. 1 | Development of INSPECT and screening for Rho GTPases and effector protein interactions. a** Scheme of Phase-separated condensates for detecting P-P interactions. Schematic was created in BioRender. Heo, W. (2025) https://BioRender.com/2718swf. **b** Condensate formation efficiency was measured by counting the number of co-localized condensate-formed cells in total co-transfected cells with FRB-CFP-FT and FKBP-RFP (mStrawberry, $n = 150$ cells, mOrange, $n = 150$ cells, dTomato, $n = 150$ cells, LanRFP, $n = 173$ cells, DsRed, $n = 393$ cells, $N = 3$ independent experiments, one-way ANOVA). Means and SD are shown. **c** Fluorescence intensity (RFP) changes before and after the addition of rapamycin (500 nM). After rapamycin addition, cytosolic dispersed FKBP-DsRed was redistributed into clusters ($n = 13$ cells, $N = 3$ independent experiments). **d** HeLa cells showing activity-dependent interaction between HRas and RBD$_{Raf1}$. Cells were co-expressed with DsRed-HRas (S17N or Q61L) and RBD$_{Raf1}$-CFP-FT. Clusters were only detected when HRas was active (HRasQ61L). Line profiles corresponding to the yellow lines indicated in merged images. The Pearson correlation coefficients (r) with corresponding two-tailed $P$-values are shown. **e** Table shows the calculation of the condensate formation efficiency (0–100%, top), signal-to-noise ratio (SNR, 0–1, middle), and protein interaction index (PI, 0–1, bottom). **f** Tables showing the Ras family and Raf1 interactions, as evaluated using condensate formation efficiency (0–100%, top), signal-to-noise ratio (SNR, 0–1, middle), and a protein interaction index (PI, 0–1, bottom). **g** heatmap of the PIs between the Rho small GTPases and effectors. For each GTPase, the PIs with the effectors within a gene family are reduced, and the maximum values are shown. Hierarchical clustering (with Euclidean distance and complete linkage) was performed for GTPases and effectors, respectively. The number of optimal clusters was determined based on pvclust (Pvclust: an R package for assessing the uncertainty in hierarchical clustering). **h** Representative images of HeLa cells expressing DsRed-RhoAQ63L (ΔCAAX), -Rac1Q61L (ΔPB and ΔCAAX), -Cdc42Q61L (ΔCAAX) with GBD$_{ROCK1}$-CFP-FT. The insets show an enlarged image of the boxed areas. The line scan profiles were obtained from the yellow lines in each image. The Pearson correlation coefficients (r) with corresponding two-tailed $P$-values are shown. Scale bars, 20 μm. **i** Table displays the values for efficiency, SNR and the product of efficiency and SNR. These raw values are then normalized by dividing each by the maximum value in the respective column to yield a scaled PI.

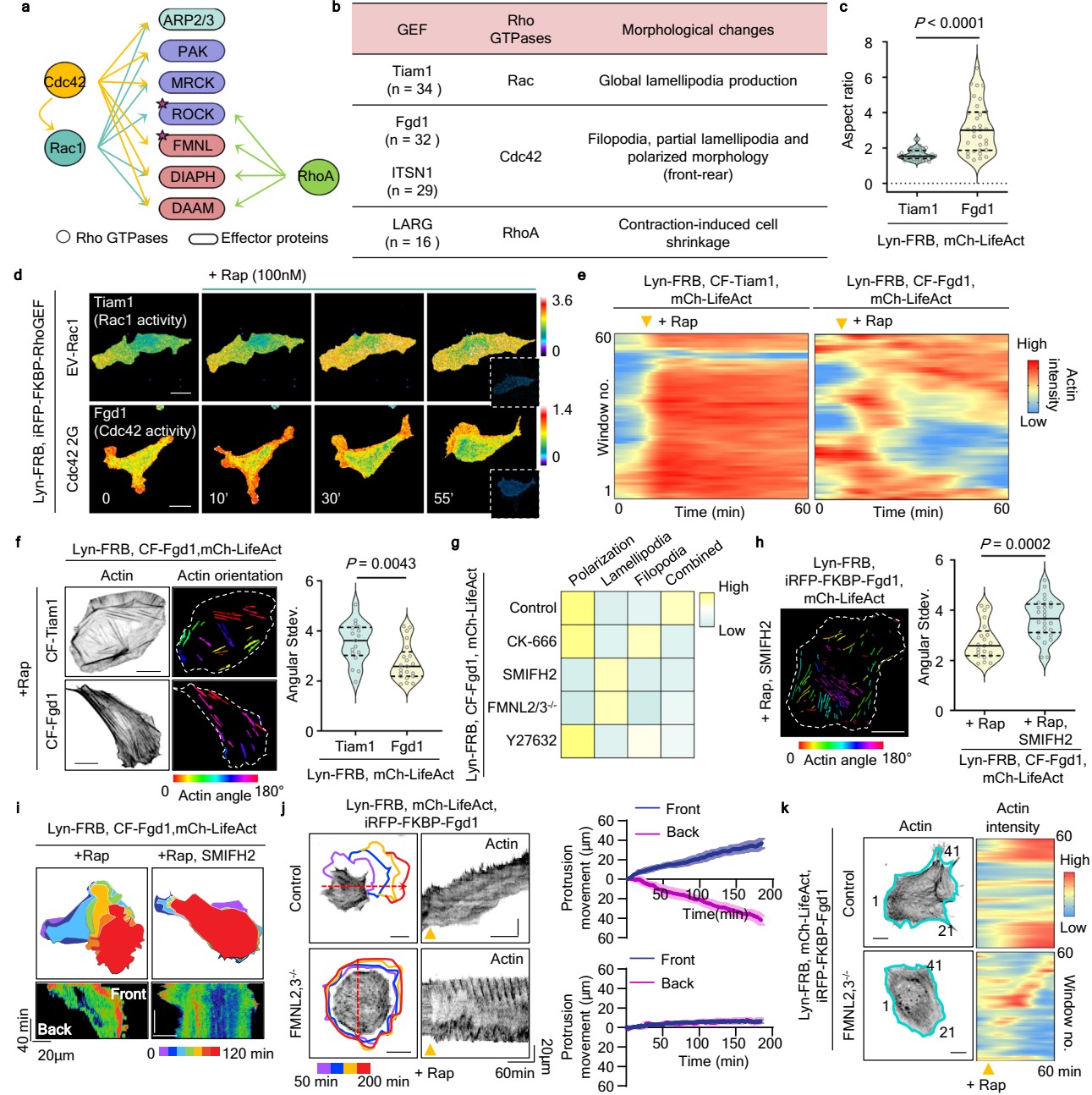

**Fig. 2 | Distinct morphological changes through cooperative cytoskeletal rearrangements by an ensemble of Rho GTPases and effectors. a** Summary of interaction profile of Rho GTPase and effector protein interactions. Schematic was created in BioRender. Heo, W. (2025) https://BioRender.com/2718swf. **b** Table shows a list of the Rac1-, Cdc42-, and RhoA-specific GEFs and the corresponding cellular morphological changes induced by each GEF. **c** Graph showing aspect ratio in NIH 3T3 cells expressing Lyn-FRB, mCh-LifeAct, CF-Tiam1 ($n = 34$ cells) or CF-Fgd1 ($n = 32$ cells). Cells were treated with 500 nM rapamycin. Median value, upper and lower quartiles (25th and 75th percentiles, dotted line) are shown.
**d** Fluorescence ratio images showing Rac1 (top) and Cdc42 (bottom) activity over time in NIH3T3 cells expressing Lyn-FRB, iRFP-FKBP-Tiam1 or Fgd1 with EV-Rac1 or Cdc42 2G FRET sensors. **e** Kymograph of actin intensity in NIH3T3 cells expressing Lyn-FRB, iRFP-FKBP-Tiam1 (left) or Fgd1 (right) with mCh-LifeAct. **f** The degree of actin fibers was calculated and colorized in a range of 0° to 180 °. Actin orientation and morphological changes induced by Rac1 (Tiam1, $n = 17$ cells) or Cdc42 (Fgd1, $n = 21$ cells) are shown. Violin plot shows the average stress fiber angle deviation in Rac1- or Cdc42-activated cells. Median value, upper and lower quartiles (25th and 75th percentiles, dotted line) are shown. Scale bars, 20 μm. **g** Heatmap shows the effect of inhibitor treatment in Cdc42-activated cells. **h** Actin orientation of NIH3T3 cells expressing Lyn-FRB, mCh-LifeAct, and iRFP-FKBP-Fgd1 treated with rapamycin and SMIFH2. Violin plot showing the average stress fiber angle deviation and F-actin intensity in NIH3T3 cells expressing Lyn-FRB, mCh-LifeAct and iRFP-FKBP-Fgd1 treated with rapamycin ($n = 21$ cells) or rapamycin and SMIFH2 ($n = 27$ cells). Median value, upper and lower quartiles (25th and 75th percentiles, dotted line) are shown. **i** A sequential overlay of colored-cell masks (top) showing changes in the morphology of cells expressing Lyn-FRB, CFP-FKBP-Fgd1, and mCh-LifeAct. The rainbow scale color bar represents specific time points over 2 h. Cells were treated with rapamycin (left), rapamycin and SMIFH2 (right). The kymograph of mCh-LifeAct shows the changes in actin intensity over time (bottom). **j** Overlay of colored masks (left) showed the cell outlines at each time point for control (H9) and FMNL2,3$^{-/-}$ cells expressing Lyn-FRB, mCh-LifeAct, and iRFP-FKBP-Fgd1 following rapamycin treatment. The kymographs (right) represent the movement of actin along with the red dotted arrows. Graph showing average protrusion and retraction at front and back of Cdc42-activated H9 ($n = 16$ cells) and FMNL2,3$^{-/-}$ cells ($n = 15$ cells). Means and SEM are shown. **k** Representative images (left) and window analysis for F-actin intensity (right) in control (H9) and FMNL2,3$^{-/-}$ cells expressing Lyn-FRB, mCh-LifeAct, and iRFP-FKBP-Fgd1. Cells were treated with rapamycin and monitored for 2 h. Cyan lines represent 60 segmented window regions along the edge. The yellow arrow indicates the time point of rapamycin addition. Scale bars, 20 μm. Unpaired two-tailed t-test $P$-values are shown in (**c**), (**f**), and (**h**).

reduced FP formation, decreased cell elongation, and increased LP formation (Supplementary Fig. 5h, i). Despite Cdc42 activation, the LP formation observed in these cells may be due to the remaining Arp2/3 or to activation of Rac1 by Cdc42[23,24].

To further investigate the molecular basis underlying the distinct roles of Rac1 and Cdc42, we examined previously reported switch-of-function mutants in which five amino acids were exchanged between the two proteins (Rac1 5 a.a. mt and Cdc42 5 a.a. mt)[25] (Supplementary Fig. 5j). We then analyzed their effector interaction profiles to understand how these sequence changes affected downstream signaling (Supplementary Fig. 5k). Notably, the Rac1 5 a.a. mutant lost its interaction with ROCK but gained binding to FMNL, whereas the Cdc42 5 a.a. mutant exhibited markedly reduced FMNL3 interaction. These changes were accompanied by reversed morphological features, including the loss of lamellipodia and increased filopodia in Rac1 5 a.a. mt-expressing cells, suggesting that these phenotypes arise from the combined gain of FMNL binding and loss of ROCK binding. These findings suggest that the effector-specific roles of Rac1-ROCK and Cdc42-FMNL interactions in shaping actin-based cell morphology.

### Cdc42-FMNL interaction regulates the establishment and reinforcement of cell polarity
We used a chemically induced dimerization (CID) system to recruit specific GEFs (Tiam1 for Rac1, Fgd1 for Cdc42, and LARG for RhoA) to the plasma membrane (PM), thereby selectively activating endogenous Rac1, RhoA and Cdc42 (Supplementary Fig. 6a)[26,27]. We observed that each Rho GTPase induced distinct morphological changes, with endogenous Cdc42 inducing front-rear polarity despite its global activation (Fig. 2b–d and Supplementary Fig. 6b, c). To investigate whether the front-rear polarity induced by endogenous Cdc42 activation is associated with its localized activity, we used FRET-based biosensors, EV-Rac1 and Cdc42 2G, to monitor Rac1 and Cdc42 activities, respectively. Upon rapamycin treatment, persistent Rac1 activation was induced throughout the cell periphery, accompanied by membrane protrusion in almost all directions. In contrast, Cdc42 activation was initially detected throughout the cell but was eventually confined to the front, corresponding to the formation of front-rear polarity (Fig. 2d, e). This result is consistent with a previous study showing Cdc42 activation-induced self-organization despite uniform external cues[28]. Furthermore, we observed distinct actin arrangements in Cdc42-activated cells compared to Rac1-activated cells, which aligned with the direction of front-rear polarization (Fig. 2f and Supplementary Fig. 6d). Additionally, global Cdc42 activation induced local actin-rich protrusions, yielding faster and more directional migration than Rac1-activated cells (Supplementary Fig. 6e, f). ITSN1,

another Cdc42-specific GEF, was used to activate endogenous Cdc42, resulting in similar morphological changes and the establishment of front-rear polarity (Supplementary Fig. 6g, h). We hypothesize that the different interaction profiles of RhoA, Rac1, and Cdc42 and their effector proteins explain this Cdc42-induced self-polarization.

Next, we investigated which effectors locally restrict Cdc42 activity to establish and reinforce front-back polarity and increase directionality. First, based on our screening results, we treated the cells with various inhibitors to perturb Cdc42-regulated effector functions (Fig. 2g and Supplementary Fig. 6i, j). Only the pan-formin inhibitor SMIFH2 disrupted Cdc42-induced polarization and the restriction of Cdc42 activity. Cells treated with SMIFH2 lost front-back polarity and exhibited decreased actin fiber alignment (Fig. 2h, i and Supplementary Fig. 6k, l). To verify the unique effector responsible for Cdc42-induced polarization, we knocked down formin family effectors acting downstream of Cdc42, including the FMNLs and DIAPHs. In the presence of Cdc42 activation, knockdown of FMNL2/3 but not DIAPH reduced cell polarity, speed, and directionality (Supplementary Figs. 7a–d and 8a–f). FMNL2/3 knockout yielded the same results (Supplementary Fig. 9a, b). Notably, FMNL2/3-deleted cells sometimes showed a transient actin-rich protrusion encircling the cell periphery without accumulating in any particular area (Fig. 2j, k, and Supplementary Fig. 9c–e). These results demonstrate that FMNL2 and FMNL3 determine the subcellular locations where sustained Cdc42 activation leads to actin polymerization, reinforcing cell polarization and directionality.

Interestingly, we expected that restricting Cdc42 activity would promote leading-edge longevity, resulting in higher directionality, but we often observed such cells making spontaneous directional changes rather than maintaining directional migration (Supplementary Fig. 6e, f). Because Cdc42 is thought to activate Rac1 at the protrusions[23,24], we hypothesized that Cdc42-induced Rac1 activity triggers repolarization. To determine whether Rac1 affects repolarization, we examined directionality changes in Rac1-deleted cells, finding that such cells showed increased directionality compared to controls (Supplementary Fig. 9f). Our data suggest Rac1 reduces directional movement by promoting repolarization and spontaneous directional changes.

### Rac-ROCK interaction regulates arc stress fiber and adhesion formation
Next, we investigated which effectors are crucial for Rac-specific functions. As expected, we observed that endogenous Rac activation caused global LP protrusion. Interestingly, Rac1-activated cells produce arc stress fibers (SF) parallel to the PM and exhibit recruitment of adhesion proteins (Supplementary Fig. 10a). The arc SF in these cells

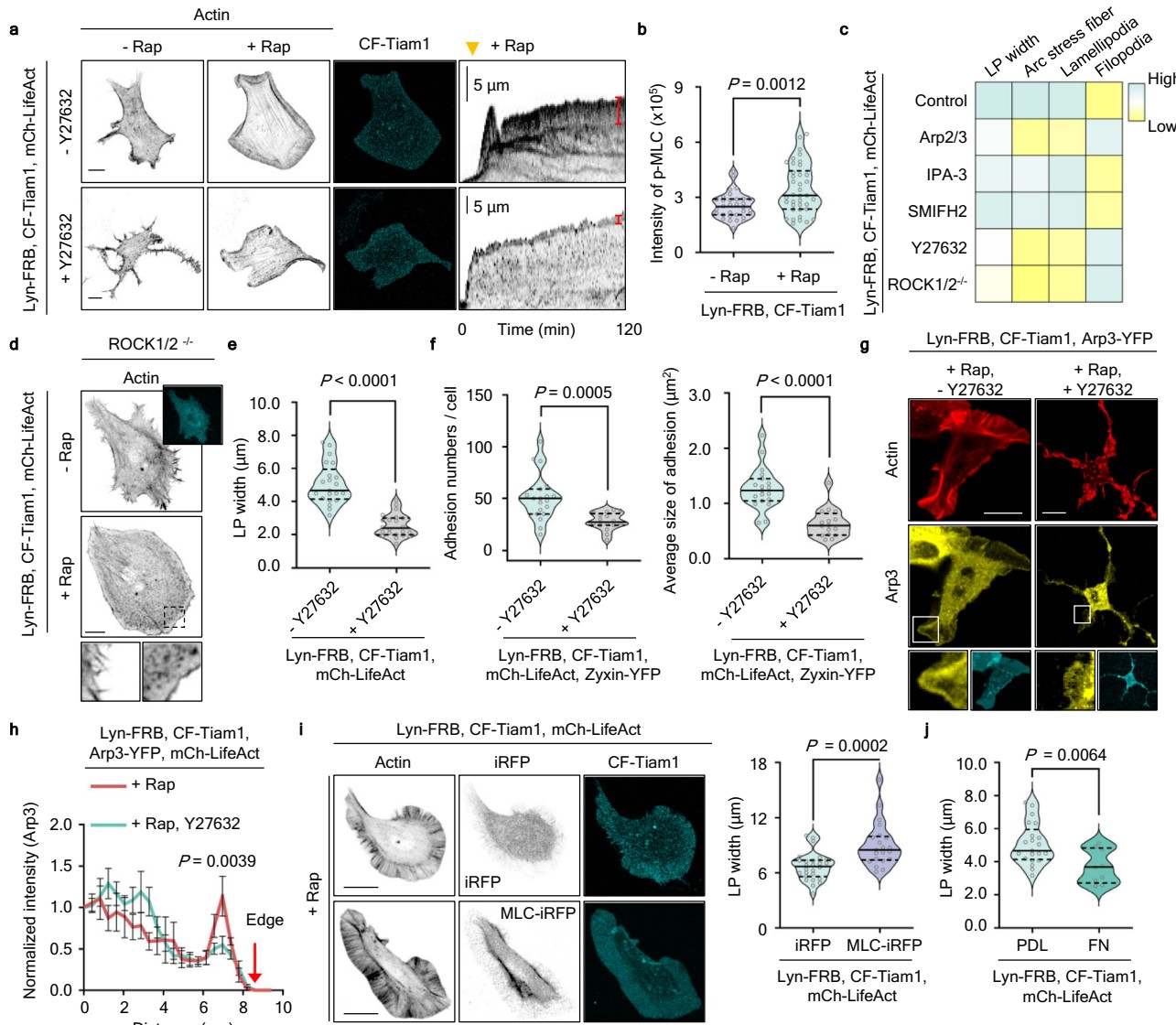

**Fig. 3 | Rac parallelly controls ROCK/MLC-mediated and Arp2/3-mediated branched actin polymerization to generate arc stress fiber. a** NIH3T3 cells expressing Lyn-FRB, CF-Tiam1, mCh-LifeAct (inverted contrast). Cells were left untreated (upper) or treated with 20 µM Y27632 (lower) for 2 h and then treated with 500 nM rapamycin. Images were captured 2 h after rapamycin treatment. Red bars indicate lamellipodia (LP). Scale bar, 20 µm. **b** Quantification of p-MLC intensity from untreated (*n* = 32 cells) or treated with rapamycin (*n* = 38 cells), obtained from *N* = 3 independent experiments. Median value, upper and lower quartiles (25th and 75th percentiles, dotted line) are shown. **c** Heatmap shows the effect of inhibitor treatment in Rac1-activated cells. **d** Representative images of ROCK1/2$^{-/-}$ cells expressing Lyn-FRB, CF-Tiam1, mCh-LifeAct (inverted contrast) before and after adding 500 nM rapamycin. Scale bar, 20 µm. **e** Graphs showing lamellipodia width, defined as the shortest vertical distance from the membrane to arc SFs, calculated from more than 20 points within a single cell and presented the mean value. Cells were treated with 500 nM rapamycin (*n* = 21 cells), with or without 20 µM Y27632 (*n* = 22 cells). Median value, upper and lower quartiles (25th and 75th percentiles, dotted line) are shown. **f** Graph showing adhesion numbers and average size in NIH 3T3 cells expressing Lyn-FRB, CF-Tiam1, Zyxin-YFP and mCh-LifeAct.

Cells were treated with 500 nM rapamycin (*n* = 19 cells), with or without 20 µM Y27632 (*n* = 16 cells). Median value, upper and lower quartiles (25th and 75th percentiles, dotted line) are shown. **g, h** NIH 3T3 cells expressing Lyn-FRB, CF-Tiam1, Arp3-YFP, and mCh-LifeAct. Cells were treated with Rap (*n* = 7 cells) or Rap with 20 µM of Y27632 (*n* = 11 cells). Arp3 enrichment at the PM was disrupted by Y27632 treatment. Scale bars, 20 µm. Graph showed the normalized mean intensity of Arp3 at the lamellipodial tip. Means and SEM are shown. Statistical significance was determined by two-way ANOVA with Sidak's multiple comparison test. **i** NIH3T3 cells expressing Lyn-FRB, CF-Tiam1, mCh-LifeAct, and iRFP682 (*n* = 26 cells) or MLC2-iRFP682 (*n* = 21 cells). Cells were treated with 500 nM rapamycin, and images were captured 2 h after rapamycin treatment. The LP width increased in MLC-expressing cells. Median value, upper and lower quartiles (25th and 75th percentiles, dotted line) are shown. **j** Graph showing lamellipodia width in NIH3T3 cells expressing Lyn-FRB, CF-Tiam1, mCh-LifeAct on PDL (*n* = 21 cells)- or FN (*n* = 9)-coated plate. Median value, upper and lower quartiles (25th and 75th percentiles, dotted line) are shown. Unpaired two-tailed t-test *P*-values are shown in (**b**), (**e**), (**f**), (**i**), and (**j**).

showed localized enrichment of ROCK (endogenous and exogenous) and phosphorylated myosin light chain (p-MLC), along with an increase in the total p-MLC levels (Fig. 3a, b and Supplementary Fig. 10b–e). Although the role of RhoA in regulating SF formation through ROCK/MLS is well-established[29,30], we treated the cells with a RhoA inhibitor (C3) to exclude the potential influence of RhoA on the

formation of arc SFs driven by Rac activation. Our result shows that RhoA inhibition did not affect the formation of arc SFs driven by Rac activation (Supplementary Fig. 10f). The RhoA-inhibited cells did, however, exhibit defects in the retrograde flow of arc SF to the cell center. We next asked whether optogenetic modulation of Rac recapitulates SF generation or disruption. When we used optogenetic tools

to achieve activation (PA-Rac1)[31] or inactivation (LARIAT)[32], we found Rac modulation is both sufficient and necessary for generating arc SFs (Supplementary Fig. 10g–j). Furthermore, we tested whether endogenous Rac activation generates arc SFs in various cell types. Consistent with our findings, endogenous Rac activation promotes LP production accompanied by arc SF formation in all tested cell types (Supplementary Fig. 11).

To identify specific effectors important for Rac-induced arc SF formation, we treated cells with various inhibitors based on our previous screening results. ROCK-depleted cells and cells treated with Arp2/3 or ROCK inhibitors did not produce arc SFs (Fig. 3c). How do Arp2/3 and ROCK affect arc SF generation? Not only are arc SFs thought to require a branched actin network as a precursor, but arc SFs are lost upon depletion of p34, a component of the Arp2/3 complex[33]. Thus, we hypothesize that arc SF generation requires both Rac-induced contractility (ROCK/MLC) and branched actin networks (Arp2/3). With ROCK disrupted, neither optogenetic nor chemogenetic activation of Rac-induced arc SF formation or stabilized LP (Fig. 3c–e and Supplementary Fig. 12a, b). Despite consistent normalized cell area compared to control cells, ROCK inhibitor treatment decreased both adhesion size and numbers (Fig. 3f and Supplementary Fig. 12c). This ROCK-dependent adhesion regulation remained consistent across multiple cell types, where ROCK inhibition, either before or after rapamycin treatment, disrupted Rac-induced adhesion formation (Supplementary Fig. 12d, e). The formation of arc SFs occurs independently of adhesion formation, whereas adhesion formation is influenced by the formation of arc SFs, suggesting a sequential regulatory relationship between these two cellular structures.

Cells treated with Y27632 showed a disruption in the Arp3 signal localized to the PM. They also showed only small transient protrusions, suggesting that Rac-induced contractility is required for Arp2/3 enrichment at the PM and LP maintenance (Fig. 3g, h). Similar to our results with CK-666-mediated Arp2/3 inhibition, myosin motor inhibition via blebbistatin (BLB) treatment also disrupted arc SFs and LP maintenance (Supplementary Fig. 12f). In contrast, MLC overexpression induced a significant increase in LP width (Fig. 3i and Supplementary Fig. 12g). On the other hand when cells plated on FN substrates, where external mechanical forces were additionally applied, there was a notable decrease in LP width (Fig. 3j). Our findings suggest that Rac proteins simultaneously activate downstream ROCK/MLC- and Arp2/3-mediated pathways, which then converge to generate arc SFs by coupling branched actin filaments with myosin-mediated contractility. Moreover, the cooperative action of actin polymerization and actomyosin contractility within the cell is also crucial for modulating the balance with the external environment.

## Rac/ROCK-interaction impairment changes actin cytoskeleton structure and adhesions

Next, we investigated whether the arc SFs induced by Rac activation trigger spontaneous directional changes as they promote cellular repolarization. One previous report found that the Rac1 effector-loop mutant Rac1F37A activates PAK but cannot interact with ROCK1, preventing it from inducing LP or focal adhesion formation[11]. When we investigated the interactions of Rac1F37AQ61L using INSPECT, we found that the F37A mutant interacted only with PAK proteins and not ROCK or any other effectors (Fig. 4a). Based on the RhoA-ROCK1 complex structure (Supplementary Fig. 13a)[34], we generated a Rac1F37W mutant that showed significantly reduced affinity for ROCKs while retaining its other interactions (Fig. 4b, c and Supplementary Fig. 13b). Using Bio-Layer Interferometry (BLI), we found that the interaction affinity of RhoA-ROCK was 1.8-fold and 5.8-fold higher than those of Rac1-ROCK and Rac1F37W-ROCK, respectively (Fig. 4d). After measuring its binding kinetics and performing a GST pull-down assay, we confirmed that the F37W mutant exhibits weaker interactions than Rac1 (Supplementary Fig. 13c). Therefore, we used the

Rac1F37W mutant to further dissect the functions of Rac1-ROCK interaction.

Expression of constitutively active Rac1F37W (Rac1F37WQ61L) in Rac1-deleted hESC-derived fibroblast cells prevented LP formation and reduced cellular area and circularity (Supplementary Fig. 13d–g). Consistent with our findings on genetic and functional disruption of ROCK, the F37W mutant also exhibited reductions in adhesion and the actin cytoskeleton (Fig. 4e–g and Supplementary Fig. 13h–l). To further investigate the effect of the Rac1F37W mutation on cell migration, we tracked the nuclei of individual cells for 6 h on fibronectin (FN)-coated plates (Fig. 4h). Although Rac1F37W and Rac1 were expressed at similar levels (Supplementary Fig. 14a), cells expressing Rac1 frequently exhibited spontaneous directional changes (orthogonal turning) (Fig. 4i and Supplementary Fig. 14b, top). In contrast, cells expressing Rac1F37W showed more persistent directional movement, fewer arc SFs, and the appearance of filopodia-like protrusions instead of lamellipodia, which resulted in reduced cell size and circularity (Fig. 4e, bottom and Supplementary Fig. 14b–d). Rac1-deleted cells showed a 2.6-fold decrease in speed compared to control cells, but Rac1 expression fully rescued this reduction (Fig. 4j). Similar trends in directionality and speed were observed in Rac1F37W-expressing cells on collagen (Col)-coated plates (Supplementary Fig. 14e). Interestingly, in the absence of ECM, cells expressing Rac1 and Rac1F37W showed similar migration speed, while Rac1F37W-expressing cells showed consistently higher directionality compared to Rac1-expressing cells. This result suggests that directionality might be an intrinsically determined characteristic independent of the extracellular environment.

## ROCK is localized to the front of the cell and correlated with spontaneous directional changes

Then, we assessed the turning angle by measuring the angles between tail retraction and newly formed protrusions and evaluated the probability with which arc SF formation and orthogonal turning coincide (Fig. 5a, b and Supplementary Fig. 14f). In this measurement, cells produce turning angles close to 0 if they move in a straight direction. We found that cells expressing Rac1F37W exhibited smaller turning angles than those expressing Rac1, also showing reduced turning and impaired arc SF formation. We further analyzed the directional persistence by calculating the directional autocorrelation[35,36]. This analysis revealed that Rac1-expressing cells showed a more rapid decorrelation compared to Rac1F37W-expressing cells, indicating that Rac1F37W-expressing cells exhibit higher directional persistence during migration (Supplementary Fig. 14g).

Notably, cells undergoing an orthogonal turn recycled pre-existing arc SFs as ventral SFs to efficiently re-establish cell polarity and support the next migratory step (Fig. 5c). Related to this, microtubules and mitochondria were excluded from the front of Rac1-expressing cells but not Rac1F37W-expressing cells (Fig. 5d and Supplementary Fig. 14h, i), suggesting that arc SFs may act as a physical barrier for organelles and cytoskeleton. We further tested whether Rac/ROCK-mediated contractility triggers orthogonal turning in different cell types. Consistent with our findings, Rac1F37W-expressing RPE-1 and MDA-MB-231 cells move straighter, accompanied by disrupted lamellipodia structures and arc SFs (Supplementary Fig. 15).

Although we showed morphological and functional changes resulting from impaired interaction between Rac-ROCK and even measured the affinity of their interaction, our data did not prove the direct regulation of Rac-ROCK. Thus, we labeled endogenous ROCK1 with a green fluorescent protein (GFP) using CRISPR/Cas9 to monitor its subcellular localization during cell migration (Supplementary Fig. 16a). Notably, when we overexpressed mScarlet-Rac1 or mScarlet-Rac1F37W in GFP-tagged ROCK cells, we observed strong recruitment of endogenous ROCK to the nucleus by Rac1 compared to the Rac1F37W (Supplementary Fig. 16b). In the absence of Rac1 or Rac1F37W overexpression, we did not observe such strong nuclear localization of ROCK1. This result

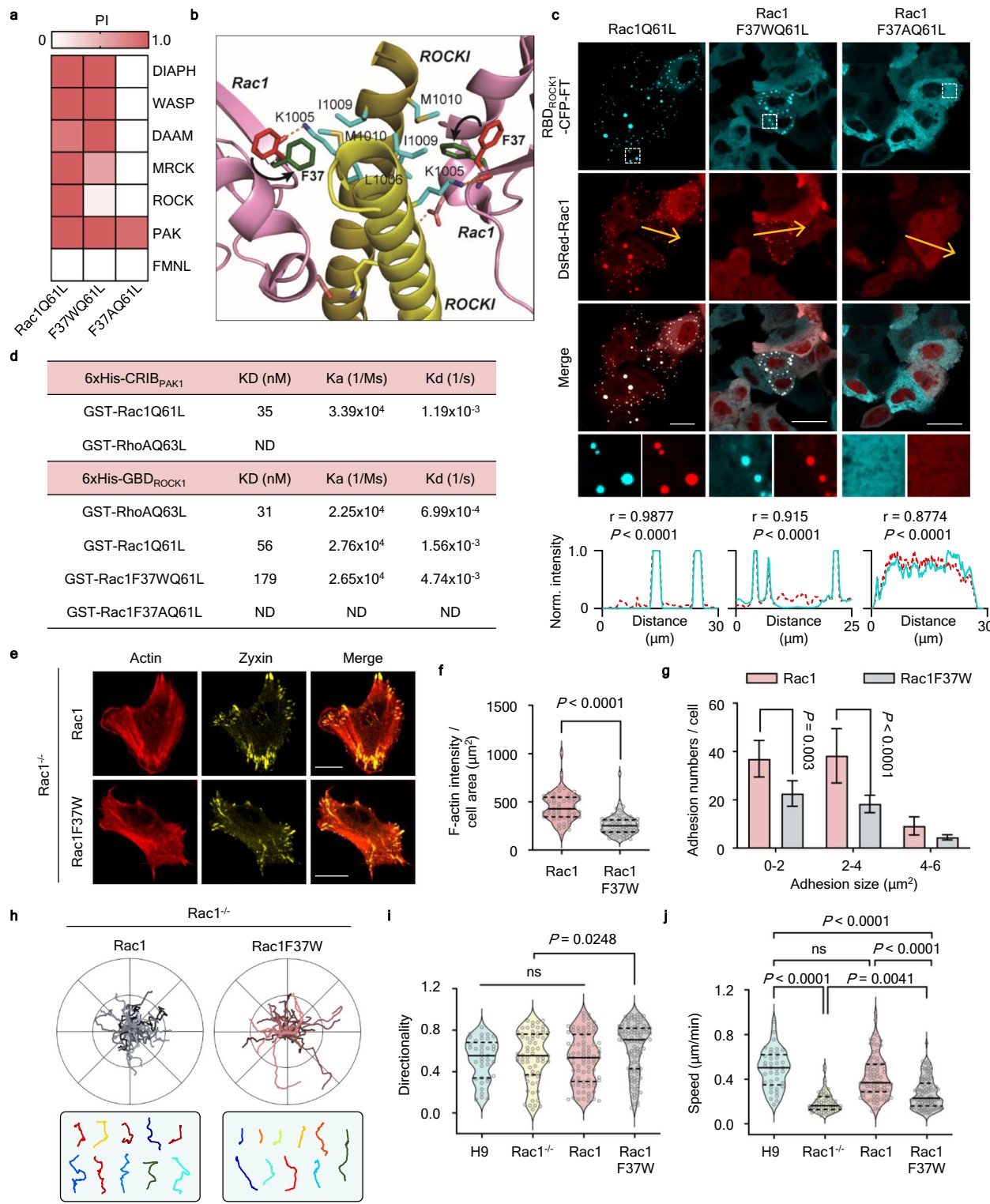

supports our data (Fig. 4a, d) and consistently demonstrates that Rac1F37W has impaired interaction with ROCK than Rac1.

To monitor a normalized spatial ROCK1 density, we divided the GFP signal by the iRFP signal (Fig. 5e). We monitored and categorized intracellular ROCK localization into four types (Fig. 5f). Consistent with the previous reports, we observed strong enrichment of ROCK at the PM during rear and protrusion retraction. When cells were changing direction, we observed strong ROCK enrichment near protrusions over a wide area spanning ~15 μm from the PM to the arc SF (Fig. 5f, g). In particular, we found a correlation between spontaneous directional

changes and higher ROCK intensity at the frontal protrusion across multiple cells (Fig. 5h, i and Supplementary Fig. 16c–f). These findings are consistent with the hypothesis that arc SF induced by Rac-ROCK interaction provides an intrinsic structural mechanism that breaks polarity and induces orthogonal turning.

## The role of Rac-ROCK-mediated contractility in modulating speed in response to environmental stimuli

We analyzed the cell migration speed in cells expressing Rac1 and Rac1F37W plated on different concentrations of FN (1, 5, and

**Fig. 4 | Impairment of Rac and ROCK interaction changes actin cytoskeleton and adhesions. a** Interaction screening results between Rac1Q61L and Rac1Q61L mt (F37A, F37W) with effector proteins. **b** Rac1–ROCKI model based on the structure of the RhoA-ROCKI complex (PDB code:1S1C) and Rac1 (PDB code: 3T5H). F37 in Rac1 may dislocate or flip (shown in black arrows) upon binding to ROCK. F37 in the Rac1-alone structure is shown in red, whereas repositioned F37 is shown in green. **c** Representative images of HeLa cells expressing DsRed-Rac1 (Q61L, F37WQ61L, F37AQ61L) with GBD$_{ROCK1}$-CFP-FT. Boxed areas are shown as enlarged images. Line scan profiles of normalized intensity corresponding to the yellow lines indicated in each image. The Pearson correlation coefficients (r) with corresponding two-tailed *P*-values are shown. Scale bars, 20 μm. **d** Binding kinetics were assessed by bio-layer interferometry (BLItz). Table summarizing the affinity (KD), association (Ka), and dissociation (Kd) constants determined by BLI between the indicated GST-conjugated Rho small GTPases and His-tagged effector proteins. **e** Rac1$^{-/-}$ cells expressing mCh-LifeAct, Zyxin-YFP, and CFP-Rac1 or CFP-Rac1F37W.

Scale bars, 20 μm. **f** Quantification of F-actin intensity in Rac1$^{-/-}$ cells expressing CFP-Rac1 (*n* = 55 cells) or CFP-Rac1F37W (*n* = 63 cells). Median value, upper and lower quartiles (25th and 75th percentiles, dotted line) are shown. *P*-value was calculated using an unpaired two-tailed t-test. **g** Distribution of average size of adhesion in Rac1$^{-/-}$ cells expressing either CFP-Rac1 (*n* = 30 cells) or CFP-Rac1F37W (*n* = 32 cells) on FN (1 μg/ml). Means and SEM are shown. Statistical significance was determined by two-way ANOVA with Sidak's multiple comparison test. **h** Cell trajectories were plotted from the origin in migrating Rac1$^{-/-}$ cells expressing either CFP-Rac1 or CFP-Rac1F37W (upper), and trajectories of 10 randomly selected cells in both cases are sown (lower). **i, j** Analysis of directionality and speed in migrating H9 cells (*n* = 37 cells), Rac1$^{-/-}$ cells (*n* = 63 cells) and Rac1$^{-/-}$ cells expressing either CFP-Rac1 (*n* = 84 cells) or CFP-Rac1F37W (*n* = 99 cells) cultured on 5 μg/mL FN. Median value, upper and lower quartiles (25th and 75th percentiles, dotted line) are shown. Statistical significance was determined by one-way ANOVA with Sidak's multiple comparison test.

10 μg/mL). It is known that the concentration of extracellular adhesion molecules produces biphasic effects on cell migration speed by modulating adhesion strength: slower migration at low and high adhesion strengths but more rapid migration at intermediate strengths[37]. A biphasic response was observed in Rac1-expressing cells, whereas cells expressing Rac1F37W maintained a constant speed across all FN conditions (Fig. 6a). Additionally, a similar biphasic pattern was observed in the translocation of YAP to the nucleus (Fig. 6b). Nuclear translocation of YAP depends on stress fibers, and our results indicate that Rac1F37W-expressing cells are unable to adapt to environmental changes due to an impairment of arc SF formation. Furthermore, the adhesion numbers remained consistent in Rac1F37W-expressing cells across various FN conditions (Fig. 6c). Although arc SFs are not directly connected to adhesions, our experiments showed that Rac/ROCK-mediated contractility-induced arc SFs play a crucial role in modulating adaptation behavior to environmental changes.

Comparing the Rac1- and Rac1F37W-expressing cells, we observed that the Rac1 protein shows a distinct pattern of adhesions under FN-coating conditions (Fig. 6d). Moreover, these patterns tend to disappear as Rac1 expression levels increase. Thus, we monitored endogenous Rac1 localization using GFP-tagged Rac1 cells and found that Rac1 was prominently positioned at the termini of actin filaments (Supplementary Fig. 17a, b). Furthermore, activating endogenous Rac1 with Tiam1 led to increased co-localization at the actin termini, coinciding with the formation of lamellipodia and stress fibers (Supplementary Fig. 17c). When we overexpressed paxillin as an adhesion marker, we observed a co-localization of Rac1 and paxillin (Fig. 6e). Notably, the correlation of co-localization between Rac1 and paxillin was maintained higher at the front of the migrating cells (Fig. 6f, g). These observations imply that Rac1 is crucial in orchestrating the coupling of the extracellular matrix and intracellular signaling pathways and cytoskeletal rearrangements. Moreover, we confirmed that recruitment of Rac1 at adhesions is regulated by ROCK-dependent contractility. Upon treatment with a ROCK inhibitor, the cell showed a loss of adhesion localized Rac1, accompanied by initiating excessive protrusions triggered by the translocation of Rac1 to PM (Fig. 6h, i and Supplementary Fig. 17d). Analysis of the correlation between Rac1 intensity and edge velocity revealed that cells treated with a ROCK inhibitor showed increased correlation, suggesting that the adhesion recruitment of Rac1 serves as a regulatory function in preventing excessive protrusion (Fig. 6j). If ROCK-dependent contractility plays a significant role in tail retraction, cells treated with a ROCK inhibitor would potentially have more pronounced effects rear than the front. However, where Rac1 localizes to adhesions due to the presence of ECM, inhibition of ROCK markedly influences frontal protrusion (Fig. 6k, upper).

In contrast, Rac1F37W-expressing cells showed similar area changes but induced more protrusions and thin protrusions than Rac1-expressing cells (Fig. 6k–m). A similar morphological change was

observed in NIH3T3 cells treated with ROCK inhibitor in PDL-coated conditions, which showed dendrite-like protrusions. These results indicate that ROCK-dependent contractility tightly regulates cell protrusions by affecting Rac1 recruitment, providing a negative regulation to Rac1. Our findings reveal that the Rac-ROCK interaction plays a dual role by enhancing randomness through providing an intrinsic structural mechanism and preventing excessive protrusion formation, ultimately enabling efficient exploration of the environment (Supplementary Fig. 17e).

## Discussion

Here, we developed a method called INSPECT to systematically investigate the interactions of Rho GTPases with downstream effectors involved in cell migration. Unlike previous protein proximity- and affinity purification-based methods[38], INSPECT can visualize binary and ternary protein complexes at the single-cell level, providing simple and precise detection of PPIs in their original cellular context with high sensitivity and specificity. INSPECT provides a direct readout that amplifies fluorescence signals in self-assembled condensates, producing much lower false-positive/negative rates than the existing approaches based on indirect reporter gene expression[39,40].

Compared to previously developed clustering-based methods[15,27], INSPECT further improves specificity by conjugating bait and prey proteins to distinct multimeric scaffold complexes. This spatial separation allows clear discrimination between genuine bait–prey interactions and condensates formed through bait–bait or prey–prey self-association. In systems using a single scaffold, such homotypic interactions could be misinterpreted as positive results, leading to false positives. INSPECT avoids this by ensuring that true interactions result in co-localized mixed condensates containing both scaffold types. Additionally, INSPECT eliminates the need for chemical inducers such as rapamycin, which were required in earlier clustering-based systems[15,27]. This avoids perturbation of endogenous signaling pathways, such as mTOR, enabling PPIs to be examined under more physiological conditions. Furthermore, by reducing the number of required expression components, INSPECT simplifies experimental design, improves assay robustness, and reduces variability stemming from differential construct expression.

Although our approach requires cytosolic localization of proteins by removing membrane-targeting sequences for optimal visualization, this technical limitation does not significantly affect the detection of genuine protein-protein interactions. We observed no correlation between the size or number of condensates distribution and the underlying interaction affinities. Instead, we developed a Protein Interaction Index (PI) using condensate formation efficiency and signal-to-noise ratio (SNR), allowing INSPECT to reflect the relative binding affinities of various PPIs.

In our study, we only used the small G binding domains (GBD) to verify protein-protein interaction, and we observed a few inconsistent

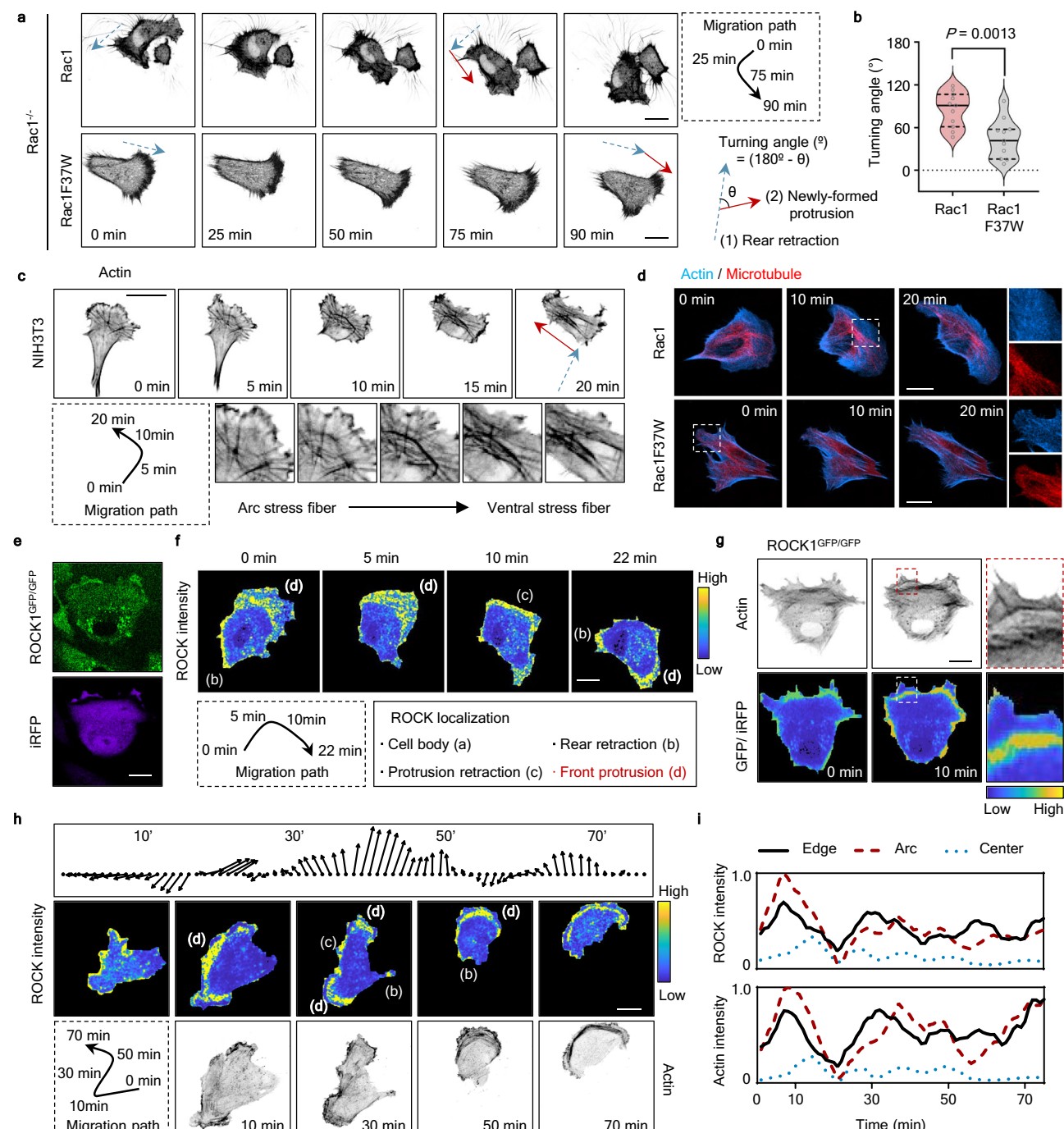

**Fig. 5 | Intrinsic Rac-regulated ROCK-dependent contractility at the front facilitates spontaneous directional changes. a** Time-lapse images of mCh-lifeAct (inverted contrast) in migrating Rac1$^{-/-}$ cells expressing CFP-Rac1 (upper) and CFP-Rac1F37W (lower). The blue arrows indicate the direction of the cell moving, where the tail retraction occurs, while the red arrows point to the direction of newly formed protrusions following tail retraction. Scale bars, 20 μm. **b** Graph showing the angle between protrusions formed after tail retraction in migrating Rac1$^{-/-}$ cells expressing CFP-Rac1 ($n = 11$ cells from three independent experiments) and CFP-Rac1F37W ($n = 11$ cells from three independent experiments). Median value, upper and lower quartiles (25th and 75th percentiles, dotted line) are shown. $P$-value was calculated using an unpaired two-tailed $t$-test. **c** Representative images of arc-to-ventral transition during orthogonal turning. Time-lapse images showed that arc stress fibers oriented perpendicular to the original direction of movement (blue dotted arrow) aligned with the direction of movement of the next round (red

arrow). Scale bar, 20 μm. **d** Representative images of Rac1$^{-/-}$ cells expressing CFP-Rac1 (upper) and CFP-Rac1F37W (lower) with iRFP682-Lifeact (blue) and FuRed-Tubulin (red). Images show that microtubules (MTs) do not penetrate but accumulate behind the arc SFs. The insets show an enlarged image of the boxed areas. Scale bar, 20 μm. **e, f** ROCK1 was tagged with GFP (ROCK1$^{GFP/GFP}$), and the ROCK1 signal was normalized by the expressed iRFP. Ratiometric images are shown on a pseudo-color range from low (blue) to high (yellow). Scale bars, 20 μm. **g** Ratiometric images of ROCK/iRFP and LifeAct images are shown at the same points. **h, i** Cell migration direction and speed with the ratiometric images of GFP (ROCK1$^{GFP/GFP}$)/iRFP and LifeAct images. The direction and length of an arrow indicate cell migration direction and speed per minute, respectively. Ratiometric images are shown on a pseudo-color range from low (blue) to high (yellow). Time-series profiles of the normalized intensity for ROCK and Actin for the defined area based on the ratiometric images in (**h**).

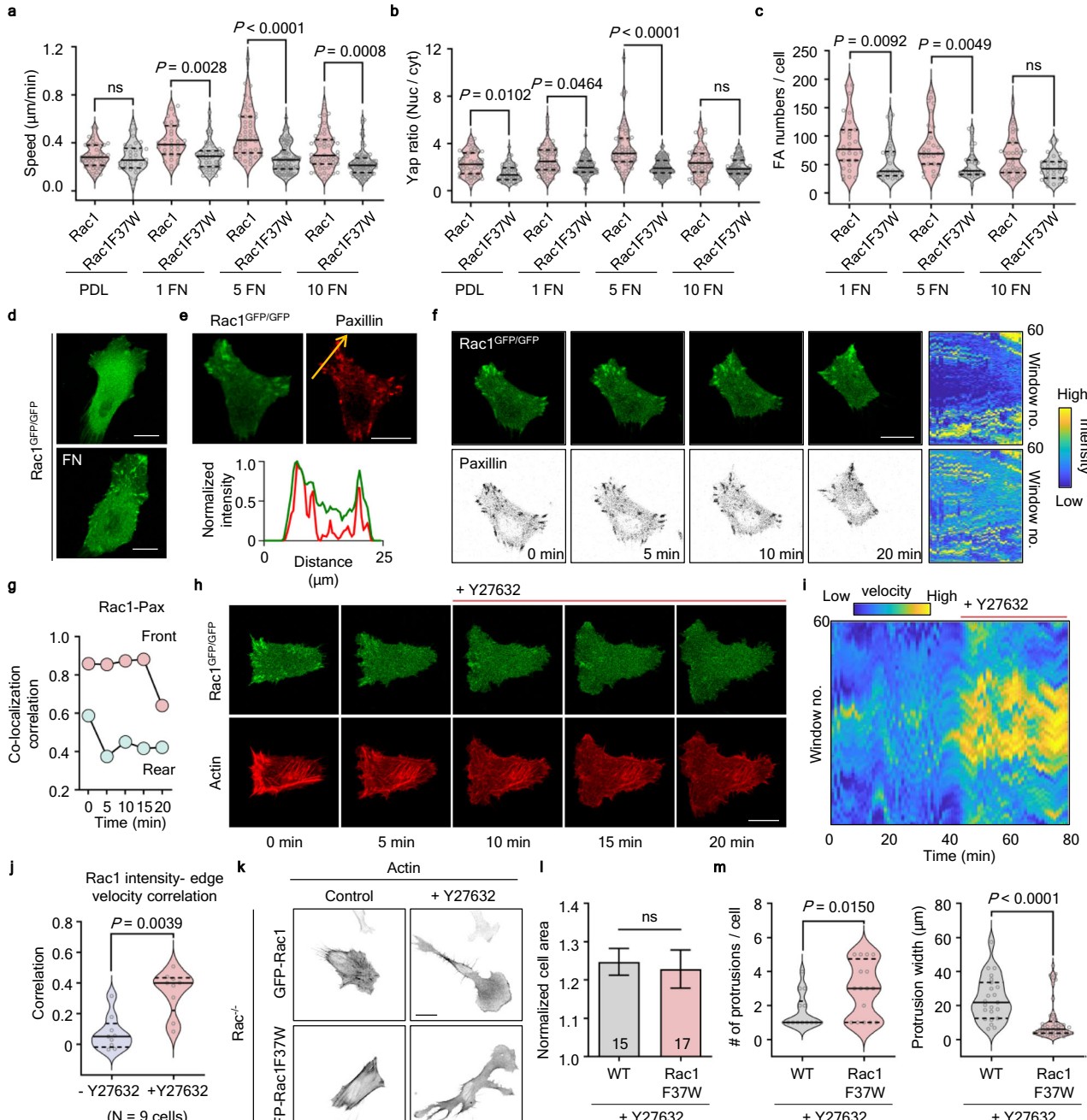

**Fig. 6 | Rac-ROCK-mediated contractility modulates speed in response to environmental changes. a**, **b** Analysis of speed and YAP nuclear translocation in migrating Rac1−/− cells expressing either CFP-Rac1 or CFP-Rac1F37W cultured on PDL, 1, 5, and 10 µg/mL FN. Median value, upper and lower quartiles (25th and 75th percentiles, dotted line) are shown. **b** Analysis of average adhesion numbers in migrating Rac1−/− cells expressing either CFP-Rac1 or CFP-Rac1F37W with Zyxin-YFP cultured on 1, 5, and 10 µg/mL FN. Median value, upper and lower quartiles (25th and 75th percentiles, dotted line) are shown. **d** Rac1 was tagged with GFP (Rac1GFP/GFP), and cells were plated on PDL- or FN-coated plates. Scale bars, 20 µm. **e** Representative images of Rac1GFP/GFP cells expressing mCh-paxillin. Images and graphs showing co-localization of Rac1 and paxillin. Scale bar, 20 µm. **f** Time-lapse images of Rac1GFP/GFP cells expressing mCh-paxillin (inverted). Window analysis for Rac1 and paxillin intensity. **g** Graph showing the co-localization correlation in panel (**f**). The migrating cell shows a higher correlation in front of cells than in the rear. **h**, **i** Representative time-lapse images and window analysis for edge velocity

in Rac1GFP/GFP cells expressing mCh-LifeAct. Cells were treated with 20 µM Y27632. **j** Graph showing the correlation between Rac1 intensity and edge velocity in Rac1GFP/GFP cells expressing mCh-LifeAct, with or without 20 µM Y27632 (n = 9 cells). Median value, upper and lower quartiles (25th and 75th percentiles, dotted line) are shown. P-value was determined by the two-tailed Wilcoxon matched-pairs signed-rank test. **k** Representative images of Rac1−/− cells expressing either CFP-Rac1 or CFP-Rac1F37W with mCh-LifeAct on FN (1 µg/ml). Cells were treated with or without 20 µM Y27632. **l**, **m** Analysis of normalized cell area, number of protrusions and protrusion width in Rac1−/− cells expressing either CFP-Rac1 (n = 15 cells) or CFP-Rac1F37W (n = 17 cells) with mCh-LifeAct on FN (1 µg/ml). Cells were treated with 20 µM Y27632. Scale bars, 20 µm. P-values were determined by the one-way ANOVA with Welch correction, followed by Games−Howell multiple comparisons test in (**a**), and by the one-way ANOVA with Turkey's post hoc test in (**b**) and (**c**). Unpaired two-tailed t-test P-values are shown in (**l**) and (**m**).

results with previously reported interactions. However, in most cases, our results are consistent with reported interactions. The inconsistency suggests the potential involvement of additional domains within effector proteins, which were not captured in our study. Using full-length effector proteins will be beneficial in addressing this observed inconsistency and enabling a more comprehensive evaluation of the interaction mechanisms. Additionally, INSPEC could be used to clarify the roles of domains and facilitate the identification of the minimal binding domains or sequences essential for interaction. We also demonstrated the specificity of the INSPECT using FK506, suggesting that INSPECT can be used as a screening platform for identifying inhibitors or competitors of protein-protein interactions. With a library of MP-fused proteins, the versatility of INSPECT could be extended to genome-wide screening and identification of key molecular mechanisms for a variety of physiological and pathological conditions.

We report that FMNL crucially contributes to establishing and maintaining cell polarity by restricting Cdc42 activity at the front of the cell in the absence of external polarized cues. In the absence of FMNLs, actin polymerization remains intact, but the decision-making processes that determine "frontness" and cell polarity are disrupted. Still, the identity of the FMNL-regulated components that activate Cdc42 and the mechanisms by which Cdc42 activity at other sites is actively or passively downregulated remain unclear. It is possible that cells use a positive feedback mechanism involving Cdc42 and FMNLs to establish and stabilize polarity for directional migration. FMNL2 is primarily localized to the Golgi apparatus, and its knockdown severely affects directed migration[41]. Cdc42-activated FMNL2 regulates actin polymerization from the Golgi, through which proteins and vesicles are anterogradely transported, thereby supporting the sustained activity of Cdc42 at the front of the cell. Furthermore, a previous report[42] documented the binding of regulatory proteins to actin filaments, suggesting a regulation mechanism for the Rho GTPases. Thus, we can not rule out the possibility of feedback mechanisms involving regulatory proteins such as GAPs, GDIs and GEFs.

Although an interaction between Rac and ROCK was previously reported[43], its functional implications in cell migration were unclear. Here, we unexpectedly revealed that the Rac-ROCK interaction contributes to the formation of arc SFs, then affecting spontaneous directional changes in various cell types. Thus, we propose that cells may employ arc SF generation in preparation for spontaneous directional changes during migration without external cues. Additionally, the arc-to-ventral transition that SFs undergo suggests the existence of an energy-saving mechanism that repurposes existing cytoskeletal architectures and allows cells to navigate the environment more efficiently.

Furthermore, Rac-ROCK enables cells to adjust their adaptation behavior in response to the complexity of the environment. While ROCK is known to act as a negative regulator of Rac1 through GAP regulation, our study demonstrates that Rac/ROCK-mediated contractility modulates Rac1 localization to control excessive activity of Rac1. By recruiting Rac1 to adhesions, cells prevent excessive protrusion formation and simultaneously, the formation of arc stress fibers driven by Rac/ROCK-mediated contractility enhances randomness in cell migration. These cooperative strategies prevent the cell from moving excessively in one direction.

Our study aimed to identify the internal mechanisms that guide cellular motility in the absence of external cues, but it was not designed to answer all questions related to directional changes. Nevertheless, our findings provide significant insights into how Rho GTPases and their effector ensembles orchestrate cell migration behavior through experiments applying chemogenetics, optogenetics, genetics, and pharmacological perturbations. Notably, we demonstrate that Cdc42-FMNL is critical for directed migration and activates Rac1, while Rac-ROCK inherently enhances randomness. Together, these mechanisms enable a balance between directional persistence and environmental exploration within integrated signaling networks, providing a cooperative strategy for regulating cell migration.

## Methods

### Ethics
All experiments complied with relevant ethical regulations. H9 (WA09) human embryonic stem cell (hESC) lines were obtained from WiCell, which has verified ethical approval for the derivation and use of this line.

### Plasmid construction
mOrange (Addgene plasmid no. 29723), mStrawberry (Addgene plasmid no. 21076), dTomato (Addgene plasmid no. 18725), and LanRFP (Genescript) were amplified and then introduced into AgeI and BsrGI sites of EGFP-N1 vector to generate mOrange-N1, mStrawberry-N1, dTomato-N1, and LanRFP-N1. FKBP was inserted into the DsRed-N1 (Clontech), mOrange-N1, mStrawberry-N1, dTomato-N1, and LanRFP-N1 to produce FKBP-DsRed, FKBP-mOrange, FKBP-mStrawberry, FKBP-dTomato, and FKBP-LanRFP, respectively.

For generating YFP-Sos$_{564-1050}$, the sequence encoding Sos1 catalytic domain (564-1050) was PCR-amplified and inserted into YFP-C1 (Clontech) vector. The expression plasmid for GFP-RBD$_{ROCK1}$, cDNA sequence encoding RBD$_{ROCK1}$ (amino acids 934-1015) was PCR-amplified and inserted at XhoI and BamHI sites of GFP-C1 vector (Clontech). To make YFP-ROCK$_{16-553}$, the sequence encoding ROCK1 catalytic domain (6-553) was PCR-amplified and inserted into XhoI and SacII sites of the YFP-C1 (Clontech) vector. To construct DsRed-MLC2, cDNA encoding myosin light chain 2 (MLC2) was PCR-amplified and inserted into DsRed-C1 (Clontech) vector. To make DsRed-HRas (Q61L, ΔCAAX), -HRas (S17N, ΔCAAX), -KRas4A (Q69L, ΔCAAX), -KRas4B (Q61L, ΔCAAX), -RRas2 (Q71L, ΔCAAX), -RRas3 (Q71L, ΔCAAX), -RIT (Q79L, ΔCAAX), -RIN (Q78L, ΔCAAX), -RHEB (Q64L, ΔCAAX), -RalA (Q75L, ΔCAAX), -RalB (Q72L, ΔCAAX), -Rap1A (G12V, ΔCAAX), -Rap1B (G12V, ΔCAAX), -Rap2A (G12V, ΔCAAX), -Rap2B (G12V, ΔCAAX), -Rerg (Q64L, ΔCAAX), -RagA (Q66L, ΔCAAX), -RagB (Q99L, ΔCAAX), -RagC (Q120L, ΔCAAX), -RagD (Q121L, ΔCAAX), -GEM1 (Q84L, ΔCAAX), -REM1 (P89V, ΔCAAX), -RasL12 (R29V, ΔCAAX), -DIRas2 (G16V, ΔCAAX), and -DIRas3 (A46V, ΔCAAX) expression plasmids, each Ras GTPases were inserted into XhoI and SacII sites of DsRed-C1 vector. To generate DsRed-RRas (Q87L, ΔCAAX), -RasD1 (S33V, ΔCAAX), and -RasD2 (S40V, ΔCAAX) expression plasmids, a sequence encoding each Ras small GTPases were inserted into XhoI and EcoRI sites of DsRed-C1 vector. To generate DsRed-RRas (Q87L, ΔCAAX), -RasD1 (Q61L, ΔCAAX), and -RasD2 (Q61L, ΔCAAX) expression plasmids, a sequence encoding each Ras small GTPases was inserted into XhoI and EcoRI sites of DsRed-C1 vector. For the construction of HRas (S17N, ΔCAAX)-CFP-FT and RBDRaf1-CFP-FT, HRas (S17N, ΔCAAX) and RBDRaf1 (51-131) were cloned into the CFP-FT vector. The expression plasmids for DsRed-Rac1 (Q61L, ΔPB), -Rac2 (Q61L, ΔCAAX), -Rac3 (Q61L, ΔCAAX), -RhoG (Q61L, ΔCAAX), -Cdc42h (Q61L, ΔCAAX), -TC10 (Q75L, ΔCAAX), -TCL (Q79L, ΔCAAX), -RhoA (Q63L, ΔCAAX), -RhoB (Q63L, ΔCAAX), -RhoC (Q63L, ΔCAAX), -RhoD (Q75L, ΔCAAX), -RhoH (S13V, ΔCAAX), -Rho6 (S71L, ΔCAAX), -Rho7 (A16V, ΔCAAX), and -RhoF (Q77L, ΔCAAX) were generated by inserting each Rho GTPases into EcoRI and BamHI sites of DsRed-C1 vector (Clontech). cDNA sequences encoding CRIB$_{PAK2}$ (amino acids 69-94), CRIB$_{PAK3}$ (amino acids 65-89 a.a), CRIB$_{PAK4}$ (amino acids 10-55 a.a), CRIB$_{PAK6}$ (amino acids 11-55 a.a), CRIB$_{PAK7}$ (amino acids 11-55 a.a), CRIB$_{WASP}$ (amino acids 230-287 a.a), RBD$_{MRCK\alpha}$ (amino acids 1558-1593 a.a)$_{\,}$ RBD$_{MRCK\beta}$ (amino acids 1583-1619 a.a), RBD$_{ROCK1}$ (amino acids 934-1015 a.a), RBD$_{ROCK2}$ (amino acids 964-1045 a.a), RBD$_{FMNL2}$ (amino acids 23-276 a.a)$_{\,}$ RBD$_{FMNL3}$ (amino acids 27-278 a.a), RBD$_{DAAM1}$ (amino acids 45-233 a.a), and RBD$_{DAAM2}$ (amino acids 41-229 a.a) were PCR-amplified and inserted at XhoI and BamHI sites of MCS-CFP-FT vector. To generate CRIB$_{PAK1}$-CFP-FT expression plasmid,

CRIB$_{PAK1}$ (amino acids 69-108 a.a) was PCR-amplified and inserted EcoRI and BamHI sites of MCS-CFP-FT vector. In order to create CRIBN$_{WASP}$-CFP-FT, a sequence encoding CRIB$_{NWASP}$ (amino acids 203-239 a.a) was PCR-amplified and then cloned into MCS-CFP-FT at NheI and EcoRI sites. To generate RBD$_{DIAPH1}$-CFP-FT and RBD$_{DIAPH2}$-CFP-FT, sequences encoding RBD$_{DIAPH1}$ (amino acids 63-260 a.a.), RBD$_{DIAPH2}$ (amino acids 88-272 a.a.) were PCR-amplified and inserted into NheI and EcoRI sites of MCS-CFP-FT. To make RBD$_{DIAPH3}$-CFP-FT, RBD$_{DIAPH3}$ (amino acids 118-287 a.a.) was PCR-amplified and inserted into NheI and SacII sites of the MCS-CFP-FT vector. Myc-ROCK1 has been described previously, and the construct was a gift from David Helfman. The sequencing encoding ROCK1 was PCR-amplified and ligated into the mCherry-C1 (Clontech) vector at XhoI sites to create mCherry-ROCK1. mCherry-LifeAct was constructed by replacing GFP from GFP-LifeAct to mCherry using NheI and BsrGI sites. For generating expression plasmids for CFP-Rac1F37A (Q61L), -Rac1F37W (Q61L), -Rac1F37A and -Rac1F37W, sequences encoding Rac1WT and Rac1Q61L were mutagenized using In-Fusion HD cloning kit (Clontech) through PCR-driven overlap extension using mutagenic oligos. The resulting PCR-amplified sequences were cloned into EcoRI and BamHI sites of the CFP-C1 (Clontech) vector. All oligonucleotide sequences used for cloning, mutagenesis, and PCR are listed in Supplementary Data 1. The mScarlet-Rac1 and -Rac1F37W were generated by replacing CFP with mScarlet (Addgene plasmid no. 85044) using NheI and BsrGI sites. The DsRed-Rac1 5 a.a mt and -Cdc42 5 a,a mt were generated by replacing CFP with DsRed (Clontech) using NheI and BsrGI sites. The DsRed-Rac1F37A (Q61L, ΔPB, and ΔCAAX) and −Rac1F37W (Q61L, ΔPB, and ΔCAAX) were generated by PCR and then inserted into EcoRI and BamHI sites at DsRed-C1 (Clonthech) vector. Arp3-EYFP has been described previously[44]. To make Zyxin-YFP, sequence coding zyxin was PCR-amplified containing attB sequences to create entry clones for the gateway system (Invitrogen). The coding region of zyxin was transferred into the XB-YFP expression vector using LR Clonase II (Invitrogen, 11791020). To generate Paxillin-mCherry, EGFP was excised with BamHI and NotI from Paxillin-EGFP (Addgene plasmid no. 15233) and replaced with mCherry from mCherry-N1 (Clontech) vector. For generating a destination vector based on pGEX-5X-2 (Amersham), a PCR-amplified CmR-ccdB cassette was inserted into the XhoI site of pGEX-5X-2 vector using the In-Fusion HD cloning system (Clontech, 639648). The coding regions of Rac1 (Q61L), Rac1F37A (Q61L), Rac1F37W (Q61L), and RhoA (Q63L) were PCR-amplified containing attB sequences and recombined to create entry clones using Gateway BP Clonase II enzyme mix kit (Invitrogen, 11789020) according to the manufacturer's instructions. To create the GST-tagged Rac1 (Q61L), Rac1F37A (Q61L), Rac1F37W (Q61L), and RhoA (Q63L), entry vectors were recombined into a Destination vector using Gateway LR Clonase II enzyme mix kit. For generating 6xHis-tagged CRIB$_{PAK1}$ and RBD$_{ROCK1}$, the coding region of CRIB$_{PAK1}$ (amino acids 69-108 a.a.) and RBD$_{ROCK1}$ (amino acids 934-1015 a.a.) were PCR-amplified containing attB sequence and recombined to create entry clones using Gateway BP Clonase II enzyme mix kit according to the manufacturer's instructions then entry vectors were recombined into a pDEST™17 vector using Gateway LR Clonase II enzyme mix kit.

To modulate Rac activity, we used the pTriEx-mVenus-PA-Rac1 gift from Klaus Hahn (Addgene plasmid no. 22007) and the CIBN-mCer-AD and mCitrine-PHR-Tiam1 as previously described[32]. The FusionRed-Tubulin was generated by replacing EGFP with FusionRed in the EGFP-α tubulin as described previously[32].

For chemical-inducible translocation experiments, we used the Lyn-FRB, CFP-FKBP-Tiam1, and CFP-FBKP-Fgd1 as previously described[27,28]. To generate the CFP-FKBP-ITSN1, the DHPH domain of ITSN1 (intersectin-1, 1212-1603) was PCR-amplified and inserted into the EcoRI site of CFP-FKBP expression vector by Gibson assembly. Then, iRFP sequences containing the NheI and BsrGI sites were inserted into the CFP-FKBP-ITSN1 to generate iRFP-FKBP-ITSN1. EV-Rac1[45]

and Cdc42-2G (Addgene plasmid no. 68814) biosensors were described previously.

The dsDNA donor vector for homologous recombination at the naïve ROCK1 locus is designed to have the eGFP-coding sequence between both homology arms, utilizing pHR-PHOX2B-L-2A-eGFP-R vector (PHOX2B donor vector) as its backbone. First, the eGFP-coding sequence was PCR-amplified (forward primer: 5′-gccataGCTAGCgtgagcaagggcgagg-3′, reverse primer: 5′-gccataGGCGCGCCccttgtacagctcgtccatgc-3′) and then integrated between the Nhe I and Asc I sites of the pHR-PHOX2B-L-2A-eGFP-R vector (referred to as the pHR-PHOX2B-L-eGFP-R vector). Each homology arm contains 1,414 bp upstream sequences (left arm) or 1,273 bp downstream sequences (right arm) relative to the start codon of ROCK1. The homology arms in the ROCK1 donor vector were generated by PCR of human genomic DNA in the WA09 (H9) hESC line and insertion of the PCR-amplified left homology arm (forward primer: 5′-TACCGAGCTCG-GATCCcgtaaatgggttcaacgccg-3′, reverse primer: 5′- CCTTGCTCACGC-TAGCcatgttgctgctgctgtgacaatgccctcttaccagcaccagca-3′) between the BamH I and Nhe I sites, and the right homology arm (forward primer: 5′-TGTACAAGGGGCGCGCCatgtcgactggggacagtttt-3′, reverse primer: 5′-AGATGCATGCCTCGAGcctggtcagggatgcttacaaca-3′) between the Asc I and Xho I sites, respectively, in the pHR-PHOX2B-L-eGFP-R vector (referred to as the pHR-ROCK1-L-eGFP-R vector). For the ROCK1-eGFP knokin (KI), the gRNA target sequence was chosen immediately upstream of the ROCK1 gene's start codon and was subsequently sub-cloned into the PX458 vector (hCas9/gRNA, Addgene plasmid no. 48138; a kind gift from Feng Zhang). This vector is engineered to co-express both gRNA and hCas9-2A-eGFP. The oligonucleotides for the PX458-ROCK1_KI construct are: forward 5′-CACCcaaccaacttcctccgcggt-3′, reverse 5′-AAAACaccgcggaggaagttggttg-3′. For the knockout (KO) of ROCK1, ROCK2, FMNL2, FMNL3, and Rac1, gRNA target sequences were selected to minimize off-target effects using the CRISPR Design Tool (http://crispr.mit.edu/). All are located downstream from the start codons of their respective genes. These sequences were also subcloned into the PX458 vector. The corresponding oligonucleotide for the PX458-ROCK1, PX458-ROCK2, PX458-FMNL2, PX458-FMNL3, and PX458-Rac1 constructs are: ROCK1-gRNA, forward 5′-CACCccgatttgg-gatcccgcagc-3′, reverse 5′-AAAACgctgcgggatcccaaatcgg-3′; ROCK2-gRNA, forward 5′-CACCccgggcattttcccgtcgg-3′, reverse 5′-AAAACccgacggg-gaaaatgcccgg-3′; FMNL2-gRNA, forward 5′-CACCccgccgacatgggcaacgca-3′, reverse 5′-AAAACtgcgttgcccatgtcggcgg-3′; FMNL3-gRNA, forward 5′-CACCtgccgggcggcagcaacaac-3′, reverse 5′-AAAACgttgttgctgccgcccggca-3′; Rac1-gRNA, forward 5′-CACCatttaagatacttacacagt-3′, reverse 5′-AAA-Cactgtgtaagtatcttaaat-3′. All insert sequences underwent validation via Sanger sequencing at the JHU Synthesis & Sequencing Facility.

## Cell culture and transfection

Undifferentiated H9 (WA09, WiCell) human embryonic stem cell (hESC) lines (WT and Rac1$^{-/-}$, ROCK1/2$^{-/-}$, FMNL2/3$^{-/-}$, ROCK1$^{GFP-ROCK1/GFP-ROCK1}$) were cultured on mitotically inactivated mouse embryonic fibroblasts (MEFs; Applied Stem Cell, ASF-1213) in a medium containing DMEM/F-12, HEPES (Gibco, 11330032), 20% knockout serum replacement (KSR; Gibco, 10828028), 0.1 mM Eagle's minimum essential medium-nonessential amino acids (MEM-NEAA, Gibco, 11140050), 1 mM L-glutamine (Gibco, 25030081), 55 μM β-mercaptoethanol (Gibco, 21985023) and 10 ng/mL FGF2 (R&D Systems, 4114-TC-01M) (hESC medium) as used routinely for hESC cultures[46]. All cells were maintained at 37 °C and 5% CO$_2$ in a humidified incubator. For fibroblast differentiation, culture media were changed gradually from hESC medium to a medium containing MEM α (Gluta-MAXTM Supplement, no nucleosides; Gibco, 32561037), 10% Fetal Bovine Serum (FBS, Gibco, 16000044) for 2 weeks. These cells were maintained for at least 4 weeks on 0.1% gelatin (Sigma, G1890)-coated flask in a medium containing DMEM (Gibco, 11965118), 10% FBS, and 1 mM L-glutamine. HeLa, A549 (ATCC, CCL-185), MDA-MB-231 (ATCC,

HTB-26), HT1080 (ATCC, CCL-121) and HK-2 (ATCC, CRL-2190) cells were maintained in DMEM (Gibco, 11995-073) supplemented with 10% FBS at 37 °C in a humidified 10% or 5% $CO_2$. NIH3T3 (ATCC) cell was maintained in DMEM supplemented with 10% Bovine Serum (BS, Gibco, 16170078) at 37 °C in a humidified 5% $CO_2$. hTERT-immortalized retinal pigment epithelial (RPE-1, ATCC, CRL-4000) cell was maintained in DMEM/F-12 (Gibco, 11320082) containing 10% FBS at 37 °C in a humidified 5% $CO_2$. U-2OS (ATCC, HTB-96) cell was maintained in DMEM containing 10% FBS with 1% Glutamax supplement at 37 °C in a humidified 5% $CO_2$. Cells were transfected using either a Neon transfection system (Invitrogen, MPK5000), Lipofectamine LTX (Invitrogen, 15338100) and JetPRIME® (Polyplus, 114-07) according to the manufacturer's instructions. For NIH 3T3 and HK-2 cells, two electroporation pulses of 1350 V for 10 ms and one electroporation pulse of 1250 V for 20 ms were used to improve transfection efficiency, respectively. H9, Rac1$^{-/-}$, ROCK1/2$^{-/-}$, FMNL2/3$^{-/-}$, and ROCK1$^{GFP\text{-}ROCK1/GFP\text{-}ROCK1}$ hESC-derived fibroblasts were maintained in DMEM containing 10% FBS with 1% GlutaMAX supplement at 37 °C in a humidified 5% $CO_2$. Cells were cultured in 0.1% gelatin from porcine skin (Sigma, G1890) coated flask and transfected using either a Neon transfection system or Lipofectamine LTX according to the manufacturer's instructions. Three electroporation pulses of 1350 V for 10 ms were used for hESC-derived fibroblast cells to improve transfection efficiency.

### Live-cell imaging, image processing and image analysis

Cells were imaged 24 h after transfection. Before imaging, all cells were plated on a 96-well glass-bottom plate (Brooks, MGB096-1-2-LG-L) or 96-well plate (μ-Plate 96 Well ibiTreat, ibidi GmbH, 89626). For NIH3T3 cells, wells were coated with 1–5 μg ml$^{-1}$ fibronectin (Sigma, F0895) or 1 mg ml$^{-1}$ poly-D-lysin (Sigma, p6407). Transfected NIH3T3 cells were serum-starved for 3 h before imaging. Live-cell imaging was performed using a Nikon A1R confocal microscope (Nikon Instruments) mounted onto a Nikon Eclipse Ti body equipped with a Nikon CFI Plan Apochromat VC objective (60×, numerical aperture (NA) 1.4, oil, or 20×, NA, air, Nikon Instruments) with Nikon imaging software (NIS-element AR 64-bit version 3.21, Laboratory Imaging). A chamlide TC system placed in a microscope stage maintained environmental conditions at 37 °C and 5% or 10% CO2 (Live Cell Instrument). CFP, GFP, YFP, mCherry, and iRFP682 (Alexa Fluor 647) were taken using 457-, 488-, 514-, 561-, and 640-nm lasers, respectively. The images were analyzed using Nikon imaging software (NIS-element AR 64-bit version 3.21, Laboratory Imaging), MetaMorph software (version 7.8.1.0, MDS Analytical Technologies) and Image J (Fiji) software.

To draw the kymographs and intensity plot, we used the 'Reslice' and 'Plot Profile' tools in Image J. Inverted images were modified in Image J using the 'Invert LUT' tool. For focal adhesion analysis, we defined adhesion met the following criteria using the 'Threshold' tool in Image J by analyzing the Paxillin-mCherry and Zyxin-YFP puncta: fluorescence intensity, 1200–4500 arbitrary units; size, 0.5–15 μm. The adhesions were counted, and cell areas were measured using Image J's 'Analyze Particle' tool.

Unless otherwise specified, cell migration assays were performed on 96 well glass-bottom plates pretreated with 5 μg ml$^{-1}$ fibronectin by tracking the nucleus of migrating cells using Hoechst 33342 (Sigma, 62249) through the 'track object' module of MetaMorph software. Using this module, the nucleus of cells was automatically tracked from frame to frame, with the XY position calculated over time. This data was used to analyze migration trajectory, distance, directionality, and cumulative path length.

Measured XY coordinates of cells were used to calculate directionality (displacement/total path length) and speed (total path length/time). The degree between two vectored migration paths

($\vec{a}$ and $\vec{b}$) was calculated as expressed in formula (1), implemented in a custom Visual Basic code of Microsoft Excel.

$$\theta = \text{ArcCos}\left(\frac{\vec{a} \cdot \vec{b}}{|\vec{a}||\vec{b}|}\right) \qquad (1)$$

Actin fibers were detected from the images by applying median and Gaussian filters, followed by intensity-based binarization. The remaining image objects were further thresholded (1) size between 10 and 150 pixels and (2) aspect ratio larger than 5 using a custom MATLAB function. Degrees of actin fibers were calculated using the 'Orientation' property of MATLAB function regionprops and colorized in hsv colormap in a range of 0° and 180° by custom MATLAB function. The actin intensity kymograph was measured at the cell boundary as described previously[28].

Autocorrelation analysis was calculated using trajectory data with a time-step of $\Delta t = 10$ min, as described in previous studies[35,36]. This time step was chosen to ensure reliable tracking of cell movement over time, allowing for the evaluation of directional persistence and correlation between subsequent positions.

### Condensate forming efficiency, SNR quantification, and PI measurement

The condensate formation efficiency was measured by counting the number of co-localized condensate-formed cells in total co-transfected cells with DsRed-small GTPases and effector-CFP-FT. SNR is measured by dividing the integrated intensity of the condensate signal by the total cellular fluorescence integrated intensity of effectors (CFP). Using ImageJ, the total fluorescence was determined by measuring the integrated density of the region of interest (ROI), including the condensates. To detect the signal from the condensates, the threshold was adjusted. Individual condensates were detected by the 'Analyze particles' function in ImageJ. Each image's threshold value was optimized to match the condensates observed in the FT (CFP) and DsRed (RFP) images. The sum of the integrated density of the condensate was divided by the total integrated density. PI is obtained by multiplying efficiency and effector SNR and is then normalized to the highest observed PI value, facilitating relative binding comparison among various small GTPases.

### FRAP experiment and analysis

Photobleaching experiments were performed with a circular region of interest using a consistent 80% laser power of total output with a 1 ms dwell time. Images were acquired every second, and photobleaching occurred after the 30th frame (30 s). For FRAP analysis, the bleached region was automatically detected and analyzed using the time-measurement module in NIS software. The mean fluorescence intensities at each time point were divided by the initial fluorescence intensity within that region. All FRAP recoveries were fit in GraphPad's Prism 10 using a one-phase association.

### Generation of knockout or knockin cell line

To generate knockin (KI) or knockout (KO) hESC line, H9 hESCs were nucleofected with both pHR-ROCK1-L-eGFP-R and PX458-ROCK1_KI plasmids for the ROCK1-eGFP KI line, with both PX458-ROCK1 and PX458-ROCK2 plasmids for ROCK1 and ROCK2 double KO line, with both PX458-FMNL2 and PX458-FMNL3 plasmids for FMNL2 and FMNL3 double KO line, or with PX458-Rac1 plasmid for Rac1 KO line; using NucleofectorTM 2b (AAB-1001, Lonza) according to manufacturer's instruction, utilizing the B-16 program by Lonza. Forty-eight hours post-nucleofection, eGFP-expressing cells were purified via FACS and seeded at a low density (50–150 cells per cm²) on mouse embryonic fibroblasts (MEFs; ASF-1213, Applied Stem Cell). To generate the ROCK1-eGFP KI line, eGFP-expressing cells were subjected to a second round of FACS purification after one week and subsequently seeded at

a low density (50–150 cells per cm$^2$) on MEFs. The resulting colonies were picked between days 10–14 and underwent two passages prior to genomic DNA isolation. For genomic DNA isolation, DNA from each hESC colony destined for KI line validation was extracted using the DNeasy Blood & Tissue Kit (69504, Qiagen), while DNA for KO line validation was extracted using the QuickExtract DNA Extraction Solution (QE09050, Epicentre). Both extraction processes followed the respective manufacturer's protocols. These regions were PCR amplified to validate the genomic regions surrounding each gRNA-binding site for ROCK1, ROCK2, FMNL2, FMNL3, and Rac1. The primers used were as follows: ROCK1, forward 5'-cgctagactgaagcacctcg-3', reverse 5'-gcagcgaaccagactaatgc-3'; ROCK2, forward 5'-gctgaggaccagcggac-3', reverse 5'-agaatcgtttggtctccggc-3'; FMNL2, forward 5'-actacccctccc-gaggaaaa-3', reverse 5'-agccagacaatgggttcgag-3'; FMNL3, forward 5'-agtcgggactcggggag-3', reverse 5'-tggggacttctcctaagggtc-3'; Rac1, forward 5'-agatgttcacagaagagacgtga-3', reverse 5'-aaagatggcttta-cagcaaaaca-3'). Following amplification, PCR products were purified using the QIAquick Gel Extraction Kit (28706, Qiagen) according to the manufacturer's guidelines. The purified PCR products underwent Sanger sequencing at the JHU Synthesis & Sequencing Facility. Clones manifesting bi-allelic knockin or bi-allelic nonsense mutations were further expanded and subjected to subsequent assays.

## Chemical-induced translocation system
At 24 h after transfection, cells were washed with OPTI-MEM (Thermo Fisher Scientific, 31985-070) and treated with selected concentrations of rapamycin (LC laboratories, R-5000), FK506 (Sigma, F4679), Y27632 (Sigma, Y05053), Blebbistatin (Calbiochem, 203391), CK-666 (Sigma, SML0006), C3 transferase (Cytoskeleton, CT04), and SMIFH2 (Sigma, S4826) as described in the manuscript. For competition experiments, cells were pre-incubated with FK506 for 10 min before treatment with rapamycin.

## FRET experiment
NIH3T3 cells were transfected with Rac1 or Cdc42 FRET-biosensor in combination with Lyn-FRB and CFP-FKBP-Tiam1 or CFP-FKBP-Fgd1. After 24 h of transfection, NIH3T3 cells were re-plated into a 96-well plate coated with 1 mg ml$^{-1}$ PDL. Emission ratio imaging was performed with a 457-nm excitation laser/535-nm emission YFP HYQ filter (Nikon, 96345). The response of the Rac1 or Cdc42 FRET sensor to each GEF was analyzed by Metamorph and NIS software.

## siRNA experiment
RNA interference was conducted in NIH3T3 cells using small interfering RNAs (siRNA) for FMNL2, FMNL3, DIAPH1, DIAPH2, and DIAPH3. siRNA for FMNL2 was generated referring to previous research (5'-GGAAGTCTGCGGATGAGATAT-3'). Other siRNA were purchased from OriGene Technologies (Rockville, MD). For the knockdown of target genes, NIH3T3 cells were transfected with each siRNA using Lipofectamine RNAiMax (13778-075, Invitrogen) according to the manufacturer's instructions. After 48-h incubation, cells were harvested for qRT-PCR or western blotting and transfected with DNA plasmids for live-cell imaging. Catalog numbers of siRNA specific for each gene are as follows: FMNL3 (SR421776C), DIAPH1 (SR422446B), DIAPH2 (SR422104C) and DIAPH3 (SR421265B).

## Quantitative PCR
We isolated total RNA from siRNA-transfected NIH3T3 cells on 6-well plates using PureLink RNA Mini Kit (Invitrogen, 12183018A) and reacted with a TURBO DNA-free kit (Invitrogen, AM1907) to completely remove genomic DNA. The extracted RNA was reverse transcribed by SuperScrpit$^{TM}$ III First-Strand Synthesis System (Invitrogen, 18080-051) and quantified using SYBR Green Realtime PCR Master Mix (TOYOBO, QPK-201T) and Real-time PCR detection system (Bio-Rad, CFX96). qPCR primers were designed by IDT primer quest tool and listed as follows:

GAPDH-F (5'-CAACTTTGGCATTGTGGAAGG-3'), GAPDH-R (5'-GTGGATGC AGGGATGATGTT-3'); DIAPH1-F (5'-CGAAGTCCTCAAGTTICCTGAT-3'); DIAPH1-R (5'-CTAAGCTCTTCTGCAGGTTCTC-3'); DIAPH2-F (5'-TGTAG-CAATGGAACAAAGCATTC-3'); DIAPH2-R (5'-TTCTCTGGCATTCTGGG-TAAAG-3'); DIAPH3-F (5'-CGAAAGAGCGAGCAGAGAAA-3'); DIAPH3-R (5'-AGCAGACTATCCATCACTCCT-3'). The relative expression level was determined by the $2^{\Delta Ct}$, where $\Delta Ct = Ct$ (sample) $- Ct$ (GAPDH).

## Photoactivation
Photoactivation was conducted with a 488-nm laser emitted through a Galvano scanner incorporated in a hybrid confocal scan head with a high-speed hyper selector (Nikon). Using the photostimulation module in NIS software, we adjusted the shape and size of the photoactivated region. A laser power of 6.5 µW (measured with an optical power meter from ADCMT) was used for photoexcitation. In the case of the PA-Rac1 experiment, photostimulation was given at 15-s intervals. For LARIAT experiments, photostimulation was given with 3 loops of 0.5-s stimuli every 4 min.

## Immunofluorescence and phalloidin imaging
NIH3T3 cells expressing Lyn-FRB, CF-Tiam1, and mCherry-LifeAct were treated with 500 nM rapamycin. After 2 h, cells were fixed with PBS containing 4% formaldehyde for 15 min at room temperature. Cells were blocked and permeated with PBS containing 0.1% TritonX-100 and 5% normal goat serum (Abcam, ab7481) for 1 h. Primary antibody, Rabbit anti-ROCK1 monoclonal antibody (1:500, sc-5560, Santacruz), anti-ROCK2 (1:500, Santacruz, sc-1851) and phosphor-MLC2 (1:300, Cell signaling, 3671) were treated in PBS containing 0.1% Tween20 at 4 °C overnight. After washing three times at room temperature with 0.1% Tween20 in PBS (PBST), cells were incubated with Alexa Fluor 647-conjugated goat anti-rabbit IgG (1:1000, Thermo Fisher Scientific, A-21244) in PBST for 1 h at RT. Alexa Fluor 594 Phalloidin was diluted into PBS (1:200, Invitrogen, A12381) and incubated for 1 h at RT. Samples were washed three times with PBS, and the image was acquired using a Nikon A1R confocal microscope (Nikon Instruments).

## Immunoblot analysis
Whole-cell lysates were prepared using PRO-PREP solution (iNtRON Biotechnology, 17081). Protein in lysates (15 µg - 30 µg of total protein per sample) were resolved by sodium dodecyl sulfate-polyacrylamide gel electrophoresis (SDS-PAGE) on a NuPAGE Novex 4–12% Bis-Tris gel (Invitrogen, NP0321). According to the manufacturer's instructions, proteins were transferred to a nitrocellulose membrane using an iBlot Transfer Stack and an iBlot Gel Transfer device (Invitrogen, IB1001). After the transfer process, membrane blocking was performed using Odyssey blocking buffer (LI-COR Biosciences, 927-40000) for 1 h. The blocked membranes were incubated primary antibody diluted blocking buffer and then incubated with the secondary antibody. Blots were then scanned with an Odyssey CLx Infrared Imaging system (LI-COR Biosciences, P/N 9140-WP). Primary and secondary antibodies used in western blot as follows: rabbit anti-GAPDH (1:500, Santa Cruz, sc-51907), rabbit anti-phosphor-MLC2 (1:300, Cell signaling, 3671), FMNL2 (cross-reactive to the FMNL3; 1:500, Abcam, ab57963), goat anti-rabbit IRDye 680RD (1: 10,000, LI-COR, 926-68071).

## Protein expression, extraction and pull-down assay
Overnight cultures of E. coli BL21 (DE3) carrying each expression plasmid, including GST-tagged Rho GTPases (-Rac1Q61L, -Rac1-F37AQ61L, -Rac1F37WQ61L, and −RhoAQ63L), were subcultured 1:1000 and grown with shaking at 37 °C to an OD600 of 0.5. Protein expression was induced by incubation with 0.1 mM isopropyl β-D-1-thiogalactopyranoside (IPTG, Santa Cruz, sc-202185) for 4 h at 30 °C. Cells were harvested in lysis buffer (50 mM Tris-Cl pH 8.0, 150 mM NaCl, 0.5% NP-40) containing a protease inhibitor cocktail and lysed by sonication. After lysis, debris was removed by centrifugation at

$14,000 \times g$ (4 °C). GST-tagged Rho GTPases were isolated using Glutathione Sepharose 4B (Cytiva, GE17-0756-05). The binding between GST-tagged Rho GTPases and $RBD_{ROCK1}$ was tested by transfecting NIH3T3 cells with GFP-$RBD_{ROCK1}$ using the Neon transfection system and then plating cells on 6-well plates (SPL, 30006). Whole-cell lysates, prepared using PRO-PREP solution, were centrifuged for 10 min at 14,000 rpm and 4 °C. The cleared supernatant was incubated with GST-tagged Rho GTPases in the presence of Glutathione Sepharose 4B beads at 4 °C for 2 h with rotation. Beads were washed three times in lysis buffer (50 mM Tris-Cl pH 8.0, 150 mM NaCl) and resuspended in 20 μl SDS sample buffer. GFP-$RBD_{ROCK1}$ binding was determined by Western blot analysis using rabbit anti-GFP (1:300, Santa Cruz, sc-8334), mouse anti-GST (1:300, Santa Cruz, sc-138), and rabbit anti-GAPDH (1:500, Santa Cruz) primary antibodies. This was followed by incubation of membranes with goat anti-rabbit IRDye 680RD (1: 10,000; LI-COR) and goat anti-mouse IRDye 800CW (1: 10,000; LI-COR, 926-322210) secondary antibodies. Blots were then scanned with an Odyssey CLx Infrared Imaging system.

### Biolyaer interferometry (BLI)
The KD values of the protein-protein interaction were obtained using the BLItz instrument (ForteBio). His tag-conjugated protein was loaded onto a nickel-nitrilotriacetic acid (Ni-NTA) biosensor for 90 s. The baseline was determined by incubating the sensor with 400 μl buffer for 30 s. A 4 μl volume of 6xHis-conjugated proteins was used to measure the protein association and dissociation for 120 s. BLItz Pro software (FroteBio) deduced binding kinetics using three different concentrations of proteins.

### ROCK localization analysis
In order to analyze the ROCK (or actin) intensity and the migration behavior of a cell from time-series cell microscopy images, we performed the following steps: Image segmentation: We first performed image segmentation using the fluorescence microscopy image. To minimize noise, we applied a median filter and binarized the image based on the mean intensity of non-zero pixels. We smoothened the cell boundary with dilation and erosion filters. Then, we left the largest object and removed the remains. Defining cell membrane regions: For each boundary pixel, we fitted a quadratic function using the four neighboring boundary pixels, including itself. Next, we obtained the perpendicular line to the tangent line of the quadratic function. Then, we defined the membrane regions from the boundary pixel to five consecutive pixels along the perpendicular line. Computing the cell migration speed and direction: For each cell, let its centroid position at time $t$ be denoted as $C_t = (x_t, y_t)$, then a cell speed at time $t$ is defined as follows:

$$Speed(t) = \frac{\sum_{k=-2}^{3} \sqrt{(x_{t+k} - x_{t+k-1})^2 + (y_{t+k} - y_{t+k-1})^2}}{6}$$

And the moving direction of the cell at time $t$ is given by:

$$Direction(x_t, y_t) = \frac{\sum_{k=-2}^{3} (C_{t+k} - C_{t+k-1})}{6}$$

Measuring the molecular intensity of ROCK and Actin: For each time point, we defined a protrusive region of a cell corresponding to a 45-degree angle to the left and right of the direction of the cell (90 degrees in total), then divided the region into three subregions, by the distance from the cell surface to the center: 0–10 percent, 10–50 percent, 50–100 percent as leading edge, arc area and center, respectively.

### Statistics and reproducibility
All experiments were performed with at least three biologically independent replicates, and results are presented as mean ± S.D or S.E.M.

Specific $P$-values are indicated in the figures. Data visualization, graphing, and statistical analyses were carried out using GraphPad Prism 10. No data were excluded from the analyses. Cells were plated under identical conditions, and fields for imaging were chosen at random. Experimental groups were assigned based on treatment (e.g., untreated versus treated). Blinding was not applicable, as all replicates were prepared and analyzed uniformly by the same researchers. All micrographs shown in this study are representative images of experiments repeated at least three times.

### Reporting summary
Further information on research design is available in the Nature Portfolio Reporting Summary linked to this article.

### Data availability
All data supporting the findings of this study are provided in the main manuscript and Supplementary Information. Raw confocal microscopy images are not publicly available due to file size and format limitations, but are available from the corresponding author upon reasonable request. Source data are provided with this paper.

### Code availability
The custom codes used in this study are not deposited in a public repository due to management considerations, but are available from the corresponding author upon reasonable request.

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

## Acknowledgements

We thank the members of the Heo laboratory for their helpful comments and discussions. We also sincerely thank Dr. Heewon Yang of Columbia University for his critical reading and comments. Figures 1a, 2a, Supplementary Figs. 2a, 4c, 5j and 17e were created with BioRender.com. Created in BioRender. Heo, W. (2025) https://BioRender.com/2718swf. This work was supported by the ASTRA Project through the National Research Foundation (NRF) funded by the Ministry of Science and ICT (RS-2024-00439379) and Samsung Science and Technology Foundation under Project Number SSTF-BA1902-06, Republic of Korea, for WDH. This work was also supported by the National Research Foundation of Korea (NRF) grants funded by the Korea Government, the Ministry of Science and ICT (RS-2023-NR077224 and RS-2024-00405360) for KHC.

## Author contributions

Conceptualization: H.L., S.L., W.D.H.; Methodology: H.L., Y.S., S.L., D.K., Y.O., Y.H., J.J., H.K., Y.J.L., H.M.K., G.L.; Investigation: H.L., Y.S., D.K.; Visualization: H.L., Y.S., S.L., D.K., B.H.; Funding acquisition: K.-H.C., W.D.H.; Supervision: G.L., K.-H.C., W.D.H.; Writing-original draft: H.L.; Writing-review & editing: H.L., S.L., D.K., K.-H.C., W.D.H., with input from all authors.

## Competing interests

The authors declare no competing interests.
