## [Transparent Peer Review file · Nature Communications]

A Rho GTPase-effector ensemble governs cell migration behavior

Corresponding Author: Professor Won Do Heo

Version 0:

Reviewer comments:

Reviewer #1

(Remarks to the Author)

The article by Lee et al. is aimed at exploring the interaction between Rho family proteins and the effector proteins and inferring the mechanisms of their influence on controlling cell shape and migration. They describe a method, referred to as INSPECT, aimed at detecting these interactions and evaluating their relative strengths. The authors infer interactions networks and attempt to evaluate their function in live cells.

In spite of the overall an interesting thrust of the research topic and many datasets that are well presented, the article does not make a sufficient enough advance to be suitable for publication. Below, I provide the reason for this evaluation through several specific critiques and overall conclusion.

The INSPECT methodology is not well explained and important questions and concern are raised by the schematic presentation of the method. For instance, it is not clear why rapamycin mediated multimerization would not override the weaker interactions between proteins such as small GTPases and their effectors. Rapamycin based dimerization they use is of extremely high affinity and is very slowly reversible. This method is not tested in an in vitro setting (not in cells) to validate (as is common for such methods). Even if this method worked as intended, in the cell it likely reflected the local concentrations of potentially interacting molecules rather than necessarily the affinity of the interaction, which is not discussed. The location, mobility and the size of the condensate particles are not characterized and explored. Overall, this method is likely to have a high number of false positive and negative findings, and should be characterized much more in depth.

The choice of the model system — the embryonic stem cells — is not justified. It is definitely a poor model for any physiological and developmental cell migration analysis, for which multiple other studies have been conducted and could provide positive controls. The significance of embryonic cell migration is not clear and the regulation of cytoskeleton in these cells may be very different from other cell types. Overall, one would expect more than a single cell type in such a study, as the conclusions, as presented are very general.

The discrepancy of the findings with the well established results for motile cells of various types should also serve as a warning sign about their technique and the chosen cell type. In particular, it is indeed very surprising that inhibition of RhoA in their hands did not lead to perturbation of stress fibers. There are other similar 'surprising' results that do raise further doubts about their particular techniques. They need need to be validated using orthogonal techniques and in other cell lines. Overall, there is little connection between the part of the paper describing the INSPECT results and the rest of the results, which are essentially a conventional analysis of the roles of Rho family proteins in cell cytoskeleton and cell motility regulation. Overall, there is little novelty with respect to the INSPECT input into these studies. For instance, the FMNL and Cdc42 has been demonstrated before. Therefore, the focus on this interaction in the latter part of the paper is not new.

Overall, again, the study falls short of the expected advance in terms of validation in vitro and in vivo, data interpretation, generality with respect to cell lines and the novelty of results.

Reviewer #2

(Remarks to the Author)

Lee and colleagues describe an elegant and well thought out study which sheds light on new molecular interactions that can

guide fibroblast migration in absence of external cues. The authors have developed a robust, intracellular, imaging-based screening method to determine protein-protein interactions by analyzing phase-separated condensates; this could also be a useful tool for researchers in the field. Utilizing this screening method, IMPACT, the authors have looked at multiple Rho GTPases and their possible migration-related effector proteins, and have unearthed 19 new pairs of interactions between Rac/Cdc42/Rho proteins and downstream effectors. In particular, the authors have proceeded to characterize interactions between Rac1/Cdc42 with ROCKs, FMNLs, and Arp2/3. Using a combination of chemically induced dimerization, inhibitors, and genetic mutants, the authors showed that FMNL2/3 decide the location of Cdc42 activation, which in turn, leads to actin polymerization and promotes cell polarity. It was previously known in the field that Cdc42 activity is spontaneously localized at leading edge of migrating cells and its steep gradient is thought to be responsible for cell steering and de novo polarization. However, the molecular interactions of Cdc42 that would limit its subcellular localization to maintain this gradient was not clearly understood; this study has provided a mechanistic insight to this. Next, the authors explored a surprising direct interaction between Rac1 and ROCK which they found to be responsible for arc stress fiber (SF) formation and also induced recruitment of adhesion proteins. They demonstrated that Rac simultaneously activates downstream ROCK-MLC and Arp2/3-mediated pathways which work together to form arc SFs by coupling branched actin filaments with actomyosin contractility. This interaction was shown to be vital for breaking pre-existing polarity and inducing orthogonal turning. Finally, the authors have implicated the importance of Rac-ROCK interaction in migration studies on FN-coated surfaces.

This study is highlighted by novelty in both assays used as well as its findings. Especially, the finding regarding Rac-ROCK interaction and its role in arc SF generation will be of interest to the field. Majority of data is clearly analyzed and presented with relevant statistics. The findings are laid out pretty well in the text and the transparency for the methods section is quite good. However, we have raised a few concerns which needs to be addressed experimentally before this manuscript may be accepted for publication in this journal.

Major Concerns:

The authors show that Rac-ROCK interaction regulates arc SFs and adhesion formation which is an interesting and new finding of this study. Apart from fibroblasts, did the authors check whether Rac-ROCK interaction had similar effects in other cell lines/systems? This would help with the generalizability of these findings.

Line 315-317: The authors claim that lack of response to environment in the F37W mutant is being driven by disruption of Arc SFs. However, the authors also report that F37W mutants have reduced adhesions (4e-g, 6c). How can we distinguish between phenotypes being driven by Arc SF disruption and reduced adhesions?

Minor comments:

Fig S3C and S4A: There are a lot of panels in fig S3C. To make it easier for the reader to interpret this, the authors should think of highlighting the 19 new pairs that they have reported here (for example, with a box around each). Also, it would be best if the molecular interaction map shown in fig S4A be a part of Fig S3 so that the reader could compare the analysis with the imaging data more easily in the same figure. Regarding color scheme in Fig S4A, the lightest colored (lowest SNR) 0% condensate formation efficiency dots are not visible when printed on paper; a slight darker color would help (same issue with Fig S12B).

Fig 2D and E: It would be helpful to note in the text and indicate in the figure 2D that these are FRET-based measurements. For both figures 2d and E, the authors did not comment on the 'Tiam1' part of the data; they should add the details in the main text.

Figure 2: Labels are a bit confusing. It's hard to tell what is being displayed in the image (mch-LifeAct) versus the other molecules present in the cell.

Figure 2: I'd prefer an image somewhere of recruitment of the FKBP-tagged molecule to assess the functionality of the system. The same should be done for any other figure where any recruitment is reported.

The authors should think of starting their results with data presented or explained in lines 156-171 or Figs 2B-F. This would give the reader a sense of what the broad question that the paper is trying to work towards. The sections on IMPACT development and effector screening would follow naturally.

Fig S10F: It looks like Arc SFs are going away in the Kymograph at ~80 min. Could the authors add some images of the cell around then, or add some quantification?

Fig S10H-J: Specifying "local inactivation" and "local inactivation" on the images would improve readability, especially because the label for inactivation in G (LARIAT) is not repeated in H and I anywhere.

Fig 3I: Could the authors indicate how they are measuring LP width on the image? It is difficult to see the difference

Fig 5D: Please add an image of microtubules and actin in the F37W mutant

S14B: Does this relocalization not occur in F37W overexpression? Otherwise, how do we know this is a direct interaction (line 290-91)?

There are spelling mistakes and syntax error throughout the manuscript. For example, line 200. The authors should correct them.

In the Plasmid construction section, 'DCAAX' should be replaced with 'CAAX'.

Reviewer #3

(Remarks to the Author)

Reviewer #4

(Remarks to the Author)

"I co-reviewed this manuscript with one of the reviewers who provided the listed reports. This is part of the Nature Communications initiative to facilitate training in peer review and to provide appropriate recognition for Early Career Researchers who co-review manuscripts."

Reviewer #5

(Remarks to the Author)

The manuscript by Lee and al presents an impressive study of Rho GTPase-effectors regulation in randomly migrating cells. First, the authors introduce a new method (INSPECT) to map the protein-protein interaction thanks to condensation induced by multimerization. They apply this method to map the interactions between Rho GTPases and their effectors (the supplementary figure 3c is a piece of art!). Second, they focus on two interactions (Cdc42/FMNL and Rac1/ROCK) to dissect the molecular mechanisms underlying their roles in cell polarity and cell reorientation during random migration. The amount of data and experiments is just enormous, and this manuscript, together with the rich supplementary figures, thus presents an impressive set of results. All of these results are convincing, well presented, and meaningfully discussed; I found the INSPECT method very elegant and convincing; their findings are of interest regarding the function role of Rho GTPases in cell polarity and migration and even if the reported interactions were known the authors nicely demonstrated the importance of those for directed migration and the way they are acting. To dampen a bit my opinion, the interactions that the authors are dissecting further were already known and the INSPECT method was somewhat not required for the second part of the work, but it is good that they used it again latter (for fig4a). Altogether this is a very good manuscript that deserves to be published. I only have a couple of minor questions/suggestions listed below.

For the FRAP experiment (supp fig1g-h), since there is an excess of rapamycin would it be possible to see any recovery? I expect all proteins to be trapped without any soluble fraction that could be here to participate to the turnover.

On the ternary complexes, why no condensates are formed with active RRas? Maybe it is known that RRas does not interact with Sos but it could be good to mention it here (lines 111-113).

Line 123, the authors start to look at Rho GTPases. In this section, it is not clear if the constitutively active forms are used. I think it is the case considering the work done in the next sections, but it will be good to mention it explicitly in this section. If the proteins are expressed in their inactive form, it would mean that they can be activated in the cytosol by their GEFs, which is not obvious and could be commented.

Still on this section, the fact that the CAAX motif was removed means that the interactions being probed lack the local environment of the membrane periphery, which could be very important for the fine regulation of effector binding and selectivity. I guess that the cytosolic localization is needed to see condensation, but this is a clear limitation of the method that should be discussed.

Line 151, "In contrast, FMNL2/3-deleted cells expressing constitutively 151 active Cdc42 showed reduced FP formation, elongated cellular morphology, and increased LP 152 formation (Supplementary Fig. 5h, i)". From the figure and data, the control has a more elongated phenotype compared to the mutant. Maybe this is a small mistake in the text.

In fig4i (and supp fig6i, 7d, 8f, 9b and d, 13e), there is an effect on directionality but I am not a big fan of this metric, which has many biases (eg with regards to cell speed). I suggest to use instead the persistence time, which is computed from the autocorrelation of the direction of motion, see 10.1038/nprot.2014.131. This quantification was used in 10.7554/eLife.69229 for example, a work that could further support the discussion lines 381-386.

Line 310 and fig6a, the mutant appears to still show a biphasic dependency, albeit less strong than the Rac1 wt.

Version 1:

Reviewer comments:

Reviewer #2

(Remarks to the Author)

The changes to the manuscript addressed our concerns and we now support publication.

Reviewer #3

(Remarks to the Author)

Reviewer #4

(Remarks to the Author)

Reviewer #5

(Remarks to the Author)

I thanks the authors for their revised manuscript that addresses all my comments.

Reviewer #6

(Remarks to the Author)

I have reviewed the revised manuscript as well as the reviewers' comments from the earlier round of peer review. In its current form, the manuscript remains unconvincing in three key areas: (1) the rigor of the biological findings, (2) the clarity and specificity of the biological question, and (3) the justification and novelty of the technological approach.

1. Rigor of the Biological Findings

The identification of non-canonical interaction partners of classical Rho GTPases is conceptually interesting and could contribute to our understanding of the molecular mechanisms underlying cell motility. However, the biological findings require more rigorous and physiologically relevant validation. In particular, the role of these interactions in the context of cell migration needs to be better demonstrated.

For instance, the characterization of Rac1F37W in comparison to wild-type Rac1 is informative, but the experimental system introduces interpretive complexity. Overexpression of Rac1F37W likely has dual effects, perhaps depending on the expression level—gain-of-function, and dominant-negative loss-of-function—which complicates interpretation. A more physiologically relevant approach would be to introduce the F37W mutation into the endogenous Rac1 gene and perform the relevant cell-based assays under these conditions.

Additionally, the experiments using ROCK inhibitors lack specificity. The global inhibition of all ROCK molecules is not to test the authors claim of ROCK-Rac interactions in a specific subcellular location. A more precise approach would involve selectively disrupting the interaction between ROCK and Rac, and similarly for other interactions such as CDC42 and FMNL, to provide more definitive evidence.

2. Clarity and specificity of the biological question

The manuscript states that “the intrinsic molecular machinery for spontaneous migration remains elusive,” but this question is too broad and loosely defined. Moreover, the presented data do not directly address this claim, at least not in an explicit manner. It is well-established in the field that cells generally utilize the same molecular machinery for migration regardless of external cues. If the authors propose that distinct machinery exists for cue-dependent and spontaneous migration, they should provide evidence to support this distinction.

As it stands, the study primarily focuses on Rho GTPases, indicating that the authors had a specific molecular target in mind from the outset. In that case, the central question would be better framed around the regulation of these GTPases, allowing for a clearer alignment between hypothesis and findings. Currently, there is a disconnect between the stated question and the experimental approach.

3. Justification and novelty of the technological approach

Without a clearly articulated biological question, the justification for developing a new protein–protein interaction (PPI) detection tool is weak. Given that membrane lipids and other factors such as metabolites also contribute to the regulation of cell migration, it is not evident why PPI detection is the most appropriate or necessary method for addressing the biological question.

Furthermore, the INSPECT technology appears to be a rebranding of previous work by the same group (e.g., PMID: 20133723 and 21796746). The manuscript does not clearly explain how INSPECT differs from or improves upon these earlier methods. If the modifications were essential for identifying novel Rho partners, that rationale should be explicitly stated and experimentally supported. Without this clarification, the novelty of the method remains questionable.

Version 2:

Reviewer comments:

Reviewer #6

(Remarks to the Author)

The authors have adequately addressed the concerns raised in the previous round of review.

Point-by-point revision responses

We sincerely thank the reviewers for their thoughtful and constructive feedback, which has substantially strengthened our manuscript. We have thoroughly addressed all comments and conducted the suggested additional experiments. Below is a summary table of new and revised figures incorporated into the manuscript.

Figures newly added or revised	
Fig. 1a	INSPECT screening for Rho GTPases and effector protein interactions
Fig. 2d and f	Distinct morphological changes through cooperative cytoskeletal rearrangements by an ensemble of Rho GTPases and effectors
Fig. 3a, d, g and i	Rac parallelly controls ROCK/MLC-mediated and Arp2/3-mediated branched actin polymerization to generate arc stress fiber
Fig. 5d	Intrinsic Rac-regulated ROCK-dependent contractility at the front facilitates spontaneous directional changes
Supplementary Table. 1	A list of DNA constructs used for INSPECT analysis of interactions between Rho small GTPases and effectors
Supplementary Fig. 1h	Development of INSPECT to detect and visualize protein-protein interactions using phase-separated condensates in living cells
Supplementary Fig. 3	DsRed-conjugated Rho GTPases and ferritin-conjugated effector proteins expression results
Supplementary Fig. 4a, b	INSPECT screening results for Rho GTPases and effector protein interactions
Supplementary Fig. 6a, c, d, g, i, k and l	Formin family proteins are required for Cdc42-induced symmetry breaking
Supplementary Fig. 8a	Depletion of FMNL2 and FMNL3 affects cell migration behavior induced by Cdc42 activation
Supplementary Fig. 9c, d and e	FMNL2 and FMNL3 are required for Cdc42-induced symmetry breaking through actin enrichment
Supplementary Fig. 10	Arc stress fiber generation via Rac/ROCK/MLC is recapitulated through optogenetic modulation of Rac activity
Supplementary Fig. 11	Rac/ROCK-induced arc stress fiber formation observed across various cell types
Supplementary Fig. 12a, d, e and f	Parallel activation of ROCK- and Arp2/3-mediated pathways converging in arc SF generation and regulation in lamellipodia protrusions
Supplementary Fig. 13b	Generation and characterization of Rac/ROCK interaction-impaired mutant (F37W)
Supplementary Fig. 14g	Rac/ROCK-induced contractility alters cell migration, providing an intrinsic mechanism for orthogonal turning
Supplementary Fig. 15	Rac/ROCK-induced contractility triggering orthogonal turning in different cell types
Supplementary Fig. 16b	Observation and analysis of ROCK and Rac1 localization through GFP knockin using the CRISPR/Cas9 system

Reviewer #1

We deeply appreciate Reviewer #1's insightful feedback and critical evaluation of our work. We have addressed the concerns raised and made the following changes to further improve our manuscript.

The article by Lee et al. is aimed at exploring the interaction between Rho family proteins and the effector proteins and inferring the mechanisms of their influence on controlling cell shape and migration. They describe a method, referred to as INSPECT, aimed at detecting these interactions and evaluating their relative strengths. The authors infer interactions networks and attempt to evaluate their function in live cells.

In spite of the overall an interesting thrust of the research topic and many datasets that are well presented, the article does not make a sufficient enough advance to be suitable for publication. Below, I provide the reason for this evaluation through several specific critiques and overall conclusion.

The INSPECT methodology is not well explained and important questions and concern are raised by the schematic presentation of the method. For instance, it is not clear why rapamycin mediated multimerizaiton would not override the weaker interactions between proteins such as small GTPases and their effectors. Rapamycin based dimerizairon they use is of extremely high affinity and is very slowly reversible. This method is not tested in an in vitro setting (not in cells) to validate (as is common for such methods). Even if this method worked as intended, in the cell it likely reflected the local concentrations of potentially interacting molecules rather than necessarily the affinity of the interaction, which is not discussed. The location, mobility and the size of the condensate particles are not characterized and explored. Overall, this method is likely to have a high number of false positive and negative findings, and should be characterized much more in depth.

We acknowledge the reviewer's concern regarding the use of rapamycin-mediated FKBP-FRB interaction and its potential to override the interaction between Rho small GTPases and effectors. The simultaneous depiction of FKBP-FRB and bait-prey in Fig. 1A may have caused confusion. To address this, we revised Fig. 1a, as shown in Fig. R1.

We would like to clarify that the rapamycin-induced FKBP-FRB heterodimerization system was used solely for our proof-of-concept experiments to identify ferritin partner and to characterize the specificity of ferritin-DsRed interconnection-induced condensate formation (Fig. 1b, c and Supplementary Fig. 1a-h). To detect small GTPases-effector interactions, we exclusively used DsRed-fused small GTPases and ferritin-fused effectors without rapamycin-mediated FRB-FKBP interaction. For clarity, we have included a comprehensive list of constructs in Supplementary Table 1.

We and others have demonstrated in previous reports that multivalent protein interconnections can lead to the formation of protein clusters or biomolecular condensates (PMID: 24793453, 28646204, 32066905, 27065232, 29115293, 25233328, 28041848). Based

on these findings and our live cell imaging and analyses (Fig. 1 and Supplementary Fig. 1), our observed condensates align well with previously documented protein-protein interaction-mediated assemblies.

Figure R1. Schematic diagram of direct bait and prey interaction-induced condensate formation.

Furthermore, we validated INSPECT by examining specific protein-protein interactions between 28 Ras family GTPases and Raf1 (Fig. 1f and Supplementary Fig. 1i). Our results showed complete consistency with previous reports (PMID: 7791872, 15143186, 7862125), demonstrating no false positives or negatives. Our previous study also revealed a similar approach's high fidelity (PMID: 22683270).

To establish INSPECT as a robust screening platform for comparing relative PPI strengths, we developed a protein interaction index (PI) based on condensate-forming population (efficiency) and signal-to-noise ratio (SNR) values. Importantly, we confirmed strong correlation between the PI values and protein interaction strength through Bio-layer interferometry (BLI, Fig. 4d) and GST-pulldown assay (Supplementary Fig. 13c) when examining RhoA, Rac1 and Rac1F37W with the RBD region of ROCK1. Both methods consistently showed weaker interaction of Rac1F37W compared to Rac1, aligning with our PI values (Fig. 4a).

The choice of the model system — the embryonic stem cells — is not justified. It is definitely a poor model for any physiological and developmental cell migration analysis, for which multiple other studies have been conducted and could provide positive controls. The significance of embryonic cell migration is not clear and the regulation of cytoskeleton in these cells may be very different from other cell types. Overall, one would expect more than a single cell type in such a study, as the conclusions, as presented are very general.

We thank the reviewer for this important point about model systems. Our study primarily used fibroblasts differentiated from human embryonic stem cells, not the stem cells themselves. To demonstrate the generality of our findings, we initially included mouse fibroblasts (NIH3T3,

Fig. 5c) as an additional model system. Both cell types showed consistent results regarding the role of Rho small GTPases in cell migration in the absence of external guidance cues. We have expanded our analysis to various cell lines to further address the reviewer's concern. Our new experimental results show that endogenous Rac activation induces arc stress fiber formation in diverse cell types, including A549, HT1080, hTERT-RPE-1, U-2OS, MDA-MB-231 and HK-2 cells. Moreover, we have confirmed that arc stress fibers trigger spontaneous directional changes during migration in RPE-1 and MDA-MB-231 cells, supporting the broader applicability of our findings. These new results are now included in the revised manuscript as Supplementary Fig. 11, 12 and 15.

The discrepancy of the findings with the well established results for motile cells of various types should also serve as a warning sign about their technique and the chosen cell type. In particular, it is indeed very surprising that inhibition of RhoA in their hands did not lead to perturbation of stress fibers. There are other similar 'surprising' results that do raise further doubts about their particular techniques. They need to be validated using orthogonal techniques and in other cell lines.

We appreciate the reviewer raising this critical point about RhoA perturbation. Our RhoA experiments were specifically designed to test whether arc stress fiber formation induced by endogenous Rac activation occurs independently of RhoA activity. Our results demonstrated that Rac-driven arc SF formation persists even when RhoA activity is perturbed (Supplementary Fig. 10f). We have revised the manuscript to clarify this finding (lines 210-214).

Regarding the validation of our findings, we have employed multiple complementary approaches to ensure robustness, including live-cell imaging, computational image analysis, genetic perturbations (knock-out and knock-down), chemogenetic/optogenetic tools, and structure-guided protein engineering. These orthogonal techniques, combined with our expanded cell line panel, support our conclusions across different experimental contexts.

Overall, there is little connection between the part of the paper describing the INSPECT results and the rest of the results, which are essentially a conventional analysis of the roles of Rho family proteins in cell cytoskeleton and cell motility regulation. Overall, there is little novelty with respect to the INSPECT input into these studies. For instance, the FMNL and Cdc42 has been demonstrated before. Therefore, the focus on this interaction in the latter part of the paper is not new.

Overall, again, the study falls short of the expected advance in terms of validation in vitro and in vivo, data interpretation, generality with respect to cell lines and the novelty of results.

The primary aim of our study was to analyze and compare Rho small GTPase-effector interactions comprehensively. Through INSPECT, we uncovered that despite considerable overlap in downstream effectors between Cdc42 and Rac, these proteins execute distinct functions through specific interactions with FMNL and ROCK, respectively. While the Cdc42-

FMNL interaction was previously known, our work revealed its role in reinforcing cell polarization by confining Cdc42 activity to the leading edge. Similarly, INSPECT analysis led us to discover the unique function of Rac-ROCK interaction at the cell front in promoting arc stress fiber formation and orthogonal turning.

The key contribution of INSPECT lies in enabling systematic and comparable analysis of protein-protein interactions with high sensitivity and specificity. This capability makes it a valuable tool for discovering new interactions and unraveling complex molecular networks across various biological systems.

We thank the reviewer for these constructive comments, which have helped us enhance our manuscript's quality and clarity.

Reviewer #2 (Remarks to the Author)

We sincerely thank Reviewer #2 for providing critical and constructive feedback that has substantially improved our manuscript. We have carefully addressed each point and implemented the suggested changes, as detailed below.

Lee and colleagues describe an elegant and well thought out study which sheds light on new molecular interactions that can guide fibroblast migration in absence of external cues. The authors have developed a robust, intracellular, imaging-based screening method to determine protein-protein interactions by analyzing phase-separated condensates; this could also be a useful tool for researchers in the field. Utilizing this screening method, IMPACT, the authors have looked at multiple Rho GTPases and their possible migration-related effector proteins, and have unearthed 19 new pairs of interactions between Rac/Cdc42/Rho proteins and downstream effectors. In particular, the authors have proceeded to characterize interactions between Rac1/Cdc42 with ROCKs, FMNLs, and Arp2/3. Using a combination of chemically induced dimerization, inhibitors, and genetic mutants, the authors showed that FMNL2/3 decide the location of Cdc42 activation, which in turn, leads to actin polymerization and promotes cell polarity. It was previously known in the field that Cdc42 activity is spontaneously localized at leading edge of migrating cells and its steep gradient is thought to be responsible for cell steering and de novo polarization. However, the molecular interactions of Cdc42 that would limit its subcellular localization to maintain this gradient was not clearly understood; this study has provided a mechanistic insight to this. Next, the authors explored a surprising direct interaction between Rac1 and ROCK which they found to be responsible for arc stress fiber (SF) formation and also induced recruitment of adhesion proteins. They demonstrated that Rac simultaneously activates downstream ROCK-MLC and Arp2/3-mediated pathways which work together to form arc SFs by coupling branched actin filaments with actomyosin contractility. This interaction was shown to be vital for breaking pre-existing polarity and inducing orthogonal turning. Finally, the authors have implicated the importance of Rac-ROCK interaction in migration studies on FN-coated surfaces.

This study is highlighted by novelty in both assays used as well as its findings. Especially, the finding regarding Rac-ROCK interaction and its role in arc SF generation will be of interest to the field. Majority of data is clearly analyzed and presented with relevant statistics. The findings are laid out pretty well in the text and the transparency for the methods section is quite good. However, we have raised a few concerns which needs to be addressed experimentally before this manuscript may be accepted for publication in this journal.

Major Concerns:

The authors show that Rac-ROCK interaction regulates arc SFs and adhesion formation which is an interesting and new finding of this study. Apart from fibroblasts, did the authors check whether Rac-ROCK interaction had similar effects in other cell lines/systems? This would help with the generalizability of these findings.

We thank the reviewer for raising this important point about the generality of Rac-ROCK interaction effects. To address this, we have extensively tested endogenous Rac activation-induced arc stress fiber formation in multiple cell lines, including hTERT-RPE-1 (retinal pigment epithelial), HT-1080 (fibrosarcoma), MDA-MB-231 (breast cancer), A549 (lung cancer), U-2OS (osteosarcoma) and HK-2 (proximal tubule epithelial) cells. Our results demonstrate that Rac/ROCK-mediated arc stress fiber formation occurs consistently across all tested cell lines (Supplementary Fig. 11). Using PA-Rac1, we have also obtained consistent results showing that Rac activation induces arc stress fiber formation in HUVEC cells (Fig. R2).

Figure R2. Rac1 activation-induced arc stress fiber formation in HUVEC cell. Scale bar, 20 μm .

To further validate these findings, we examined ROCK perturbation in RPE-1 and MDA-MB-231 cells, confirming that ROCK activity is essential for arc stress fiber and adhesion formation in these cell types (Supplementary Fig. 12d, e).

Furthermore, we investigated whether arc stress fibers facilitate orthogonal turning in RPE-1 and MDA-MB-231 cells. The results confirmed that arc stress fibers promote spontaneous directional changes through orthogonal turning. These new results have been included in the revised manuscript as Supplementary Fig. 11, 12d, e and 15.

Line 315-317: The authors claim that lack of response to environment in the F37W mutant is being driven by disruption of Arc SFs. However, the authors also report that F37W mutants have reduced adhesions (4e-g, 6c). How can we distinguish between phenotypes being driven by Arc SF disruption and reduced adhesions?

We thank the reviewer for this insightful comment regarding the relationship between arc stress fibers and adhesions. We confirmed that arc stress fiber formation is essential for adhesion regulation (Fig. 4E-G and 6C). This supports previous studies (PMID: 22797913, 8682874) showing that stress fibers play an important role in adhesion formation regulation. Therefore, we propose that the lack of environmental response by F37W mutant stems primarily from arc stress fiber disruption, which subsequently leads to reduced adhesions.

To further support this sequential relationship, we performed additional experiments to activate endogenous Rac to induce arc stress fiber and adhesion formation, followed by Y27632 treatment

to specifically perturb ROCK function (Supplementary Fig. 12d, e). Our results showed that disruption of arc stress fibers led to loss of adhesions, supporting our proposed mechanism. We have included these new data and revised the manuscript accordingly (lines 230-237).

Minor comments:

Fig S3C and S4A: There are a lot of panels in fig S3C. To make it easier for the reader to interpret this, the authors should think of highlighting the 19 new pairs that they have reported here (for example, with a box around each). Also, it would be best if the molecular interaction map shown in fig S4A be a part of Fig S3 so that the reader could compare the analysis with the imaging data more easily in the same figure. Regarding color scheme in Fig S4A, the lightest colored (lowest SNR) 0% condensate formation efficiency dots are not visible when printed on paper; a slight darker color would help (same issue with Fig S12B).

We thank the reviewer for these helpful suggestions to improve figure clarity. We have rearranged Supplementary Fig. 3c into supplementary Fig. 4a to facilitate direct comparison between the image data and the interaction map. We have also added yellow boxes to highlight the 19 newly identified interaction pairs in supplementary Fig. 4a. Additionally, we have adjusted the color scheme for 0% condensate formation efficiency in Supplementary Fig. 4b to ensure better visibility in print format.

Fig 2D and E: It would be helpful to note in the text and indicate in the figure 2D that these are FRET-based measurements. For both figures 2d and E, the authors did not comment on the 'Tiam1' part of the data; they should add the details in the main text.

As the reviewer suggested, we have revised the manuscript (lines 162-168) and figures to indicate that Fig. 2D and 2E present FRET-based measurements and actin intensity, respectively. We have also included a description of the result for Rac1 activation by Tiam1 in the main text.

Figure 2: Labels are a bit confusing. It's hard to tell what is being displayed in the image (mCh-LifeAct) versus the other molecules present in the cell.

As the reviewer commented, we have revised all labels in both main and supplementary figures, changing 'mCh-LifeAct' to 'actin' to improve clarity and readability.

Figure 2: I'd prefer an image somewhere of recruitment of the FKBP-tagged molecule to assess the functionality of the system. The same should be done for any other figure where any recruitment is reported.

As the reviewer suggested, we have now included images of FKBP-tagged molecules for all reported recruitment sites in our figures. Where space was limited in the main figures, we have added these images to the corresponding supplementary figures (Fig. 2d, 3 and Supplementary Fig. 6, 8a, 9, 10, 11 and 12).

The authors should think of starting their results with data presented or explained in lines 156-171 or Figs 2B-F. This would give the reader a sense of what the broad question that the paper is trying to work towards. The sections on IMPACT development and effector screening would follow naturally.

We appreciate the reviewer's thoughtful suggestion regarding manuscript organization. However, after careful consideration, we believe that introducing INSPECT earlier in the manuscript is essential, as one of our key aims was to establish INSPECT as a versatile screening platform. This organization allows us to demonstrate how INSPECT enabled the subsequent discoveries presented in the following figures.

Fig S10F: It looks like Arc SFs are going away in the Kymograph at ~80 min. Could the authors add some images of the cell around then, or add some quantification?

As the reviewer commented, we have investigated whether the apparent changes in arc stress fibers at 80 minutes represent a cell-wide phenomenon. Our analysis suggests that these changes coincide with the dynamic oscillation of lamellipodia protrusion and retraction cycles. To better illustrate this, we have added a kymograph from a second region of the cell and updated Supplementary Fig. 10f accordingly.

Fig S10H-J: Specifying "local inactivation" and "local inactivation" on the images would improve readability, especially because the label for inactivation in G (LARIAT) is not repeated in H and I anywhere.

As the reviewer commented, we have improved the readability of the figures by adding detailed labels specifying 'local inactivation (LARIAT)' and 'local activation (PA-Rac1)' to the images.

Fig 3I: Could the authors indicate how they are measuring LP width on the image? It is difficult to see the difference

We thank the reviewer for this request for clarification. For better visualization, we have replaced the representative images and added lines indicating lamellipodia width in the figures (Fig. 3a, i). To measure lamellipodia width, we calculated the shortest vertical distance from the membrane to arc stress fibers at more than 20 points within a single cell and determined the mean value. We have included these measurement details in the Supplementary Fig. 3e legend.

Fig 5D: Please add an image of microtubules and actin in the F37W mutant

As the reviewer commented, we have updated Fig. 5d to include images showing both microtubules and actin in the F37W mutant.

S14B: Does this relocalization not occur in F37W overexpression? Otherwise, how do we know this is a direct interaction (line 290-91)?

To address this question, we have tested ROCK recruitment to the nucleus in response to Rac1F37W nuclear localization. Consistent with our INSPECT (Fig. 4a) and BLI (Fig. 4d) data showing weakened interaction between Rac1F37W and ROCK, we observed reduced ROCK nuclear recruitment in Rac1F37W-expressing cells compared to Rac1-expressing cells. We have

quantified this difference by measuring nuclear ROCK intensity and included these data in Supplementary Fig. 16b and lines 310-315.

There are spelling mistakes and syntax error throughout the manuscript. For example, line 200. The authors should correct them.

We thank the reviewer for noting these errors. We have carefully proofread the entire manuscript and corrected all spelling mistakes and syntax errors.

In the Plasmid construction section, 'DCAAX' should be replaced with ' Δ CAAX'.

We have corrected 'DCAAX' to ' Δ CAAX'.

Reviewer #3 (Remarks to the Author):

We thank Reviewer #3 for their time and effort in evaluating our manuscript.

Reviewer #4 (Remarks to the Author):

"I co-reviewed this manuscript with one of the reviewers who provided the listed reports. This is part of the Nature Communications initiative to facilitate training in peer review and to provide appropriate recognition for Early Career Researchers who co-review manuscripts."

We thank Reviewer #4 for their time and effort in evaluating our manuscript.

Reviewer #5 (Remarks to the Author):

We sincerely thank Reviewer #5 for the positive assessment and constructive feedback. We have carefully addressed the comments and implemented the suggested changes to improve our manuscript, as detailed below.

The manuscript by Lee and al presents an impressive study of Rho GTPase-effectors regulation in randomly migrating cells. First, the authors introduce a new method (INSPECT) to map the protein-protein interaction thanks to condensation induced by multimerization. They apply this method to map the interactions between Rho GTPases and their effectors (the supplementary figure 3c is a piece of art!). Second, they focus on two interactions (Cdc42/FMNL and Rac1/ROCK) to dissect the molecular mechanisms underlying their roles in cell polarity and cell reorientation during random migration. The amount of data and experiments is just enormous, and this manuscript, together with the rich supplementary figures, thus presents an impressive set of results. All of these results are convincing, well presented, and meaningfully discussed; I found the INSPECT method very elegant and convincing; their findings are of interest regarding the function role of Rho GPTases in cell polarity and migration and even if the reported interactions were known the authors nicely demonstrated the importance of those for directed migration and the way they are acting. To dampen a bit my opinion, the interactions that the authors are dissecting further were already known and the INSPECT method was somewhat not required for the second part of the work, but it is good that they used it again latter (for fig4a). Altogether this is a very good manuscript that deserves to be published. I only have a couple of minor questions/suggestions listed below.

For the FRAP experiment (supp fig1g-h), since there is an excess of rapamycin would it be possible to see any recovery? I expect all proteins to be trapped without any soluble fraction that could be here to participate in the turnover.

We thank the reviewer for raising this important point about the FRAP experiment. We analyzed the fluorescence recovery profiles for FKBP-DsRed and FRB-CFP-FT (Fig. R3). Interestingly, while FKBP-DsRed showed some recovery after photobleaching (Fig. R3, left), FRB-CFP-FT exhibited reduced recovery (Fig. R3, right). This difference reflects DsRed's less extensive role in condensate formation compared to FT. This observation is supported by our finding that DsRed participates in condensate formation even at lower expression levels (0 ~ 200 a.u., Supplementary

Fig. 1e), suggesting a larger soluble fraction remains in the cytosol compared to FT.

Figure R3. Fluorescence recovery after photobleaching (FRAP) analysis of rapamycin-induced synthetic condensates of FKBP-DsRed and FRB-CFP-FT.

We performed additional FRAP experiments to explore this phenomenon further using an increased rapamycin concentration (1 μ M). These experiments showed decreased fluorescence recovery (Supplementary Fig. 1h), confirming that enhanced recruitment of FT and DsRed into condensates results in reduced recovery due to decreased background fluorescence signals.

On the ternary complexes, why no condensates are formed with active RRAs? Maybe it is known that RRAs does not interact with Sos but it could be good to mention it here (lines 111-113).

We thank the reviewer for this clarification request. We used RRAs as a negative control in this experiment, as it is known that RRAs is not required for Sos allosteric interaction with HRAs. We have added this explanation to the manuscript (lines 112-113).

Line 123, the authors start to look at Rho GTPases. In this section, it is not clear if the constitutively active forms are used. I think it is the case considering the work done in the next sections, but it will be good to mention it explicitly in this section. If the proteins are expressed in their inactive form, it would mean that they can be activated in the cytosol by their GEFs, which is not obvious and could be commented.

As the reviewer pointed out, we have revised the manuscript to explicitly state that constitutively active forms of Rho GTPases were used in these experiments (line 116 and lines 122-123).

Still on this section, the fact that the CAAX motif was removed means that the interactions being probed lack the local environment of the membrane periphery, which could be very important for the fine regulation of effector binding and selectivity. I guess that the cytosolic localization is needed to see condensation, but this is a clear limitation of the method that should be discussed.

We thank the reviewer for this insightful point regarding a methodological limitation of INSPECT. We acknowledge that removing the CAAX motif eliminates the membrane-proximal environment, which could indeed influence effector binding and selectivity. While this cytosolic localization was necessary to achieve optimal visualization of condensate formation, our validation studies

demonstrated that INSPECT detects protein-protein interactions with high fidelity, showing minimal false-positive/negative results as confirmed through comparison with previously reported interactions (Fig. 1f and Supplementary Fig. 1i). We have described this point in our revised manuscript (lines 381-383).

Line 151, "In contrast, FMNL2/3-deleted cells expressing constitutively active Cdc42 showed reduced FP formation, elongated cellular morphology, and increased LP formation (Supplementary Fig. 5h, i)". From the figure and data, the control has a more elongated phenotype compared to the mutant. Maybe this is a small mistake in the text.

We thank the reviewer for catching this error. The FMNL2/3-deleted cells expressing constitutively active Cdc42 actually showed reduced FP formation, decreased cell elongation, and increased LP formation compared to control cells. We have corrected this description in the manuscript (line 153).

In fig4i (and supp fig6i, 7d, 8f, 9b and d, 13e), there is an effect on directionality but I am not a big fan of this metric, which has many biases (eg with regards to cell speed). I suggest to use instead the persistence time, which is computed from the autocorrelation of the direction of motion, see 10.1038/nprot.2014.131. This quantification was used in 10.7554/eLife.69229 for example, a work that could further support the discussion lines 381-386.

We thank the reviewer for this helpful suggestion about migration analysis. While we initially presented speed and directionality metrics for their intuitive appeal to general readers, we agree that persistence time derived from directional autocorrelation provides a more robust measure. Following your suggestion, we have now included the autocorrelation of direction analysis in the manuscript (Supplementary Fig. 14g and lines 292-296).

Line 310 and fig6a, the mutant appears to still show a biphasic dependency, albeit less strong than the Rac1 wt.

We thank the reviewer for this careful observation. While there might appear to be a slight trend in the RacF37W mutant data, our statistical analysis showed no significant differences across the tested conditions. Although there was a marginal increase in speed on fibronectin substrates, this effect was not statistically significant, unlike the clear biphasic response observed with wild-type Rac1.

Point-by-point revision responses

We sincerely appreciate the positive evaluation of our revised manuscript by Reviewers #2, 3, 4, and 5. We are also deeply grateful to Reviewer #6 for the insightful and constructive feedback, which has significantly contributed to enhancing the rigor and clarity of our study. All comments have been carefully considered and addressed, and an additional experiment has been performed to strengthen our findings. A summary of newly added figure is provided in the table below.

Figures newly added or revised	
Supplementary Table 1.	A list of DNA constructs used for INSPECT analysis of interactions between Rho small GTPases and effectors
Supplementary Fig. 5j and k	ROCK and FMNL as key effectors transforming Rac1 and Cdc42 activities into distinct cellular morphology.

Reviewer #2 (Remarks to the Author):

The changes to the manuscript addressed our concerns and we now support publication.

- We sincerely thank Reviewer #2 for the supportive response and positive evaluation of our revised manuscript and for acknowledging the improvements made in response to the previous comments.

Reviewer #3 (Remarks to the Author):

- We appreciate Reviewer #3 for their time and effort in evaluating our manuscript and we appreciate their contribution to the review process.

Reviewer #4 (Remarks to the Author):

- We appreciate Reviewer #4 for their time and effort in evaluating our manuscript and we appreciate their contribution to the review process.

Reviewer #5 (Remarks to the Author):

I thank the authors for their revised manuscript that addresses all my comments.

- We are deeply grateful to Reviewer #5 for the kind and supportive comments throughout the review process. We sincerely appreciate the acknowledgement that our revisions have fully addressed the previous comments.

Reviewer #6 (Remarks to the Author):

I have reviewed the revised manuscript as well as the reviewers' comments from the earlier round of peer review. In its current form, the manuscript remains unconvincing in three key areas: (1) the rigor of the biological findings, (2) the clarity and specificity of the biological question, and (3) the justification and novelty of the technological approach.

1. Rigor of the Biological Findings

The identification of non-canonical interaction partners of classical Rho GTPases is conceptually interesting and could contribute to our understanding of the molecular mechanisms underlying cell motility. However, the biological findings require more rigorous and physiologically relevant validation. In particular, the role of these interactions in the context of cell migration needs to be better demonstrated.

For instance, the characterization of Rac1F37W in comparison to wild-type Rac1 is informative, but the experimental system introduces interpretive complexity. Overexpression of Rac1F37W likely has dual effects, perhaps depending on the expression level—gain-of-function, and dominant-negative loss-of-function—which complicates interpretation. A more physiologically relevant approach would be to introduce the F37W mutation into the endogenous Rac1 gene and perform the relevant cell-based assays under these conditions.

We thank the reviewer for the valuable comments and agree that introducing the F37W mutation into the endogenous Rac1 gene would provide the most physiologically relevant validation. Given the time and resource constraints of this revision, generating an endogenous Rac1F37W knock-in cell line was not feasible; we will therefore pursue it in future work.

To evaluate the physiological relevance of the F37W mutation, we had already implemented a well-controlled rescue strategy as part of our original experimental design, aiming to approximate physiological conditions as closely as possible. We performed functional comparisons of Rac1F37W and Rac1WT by exogenous expression in Rac1-knockout cells under matched expression levels, as quantified by CFP fluorescence intensity (Supplementary Fig. 14a). This controlled setup allowed a direct comparison without interference from endogenous Rac1. Additionally, migration behavior of Rac1WT-expressing cells was comparable to parental H9 cells with endogenous Rac1, confirming that Rac1WT expression fully rescues normal migration (Fig. 4i and j). These results strongly suggest that the phenotypic differences observed with Rac1F37W are due to its specific functional properties rather than expression level differences or nonspecific effects.

Moreover, in RPE-1 and MDA-MB-231 cells retaining endogenous Rac1, overexpression of Rac1F37W induced similar key phenotypes (Supplementary Fig. 11, 12, and 15)—such as disrupted arc stress fiber formation and impaired turning behavior—compared to Rac1WT, further supporting the robustness and physiological relevance of our findings.

Additionally, regarding the biological significance of the Rac-ROCK interaction in cell migration, we provide multiple lines of evidence:

1. Using a chemically inducible system, endogenous Rac activation was shown to trigger ROCK activation and subsequent arc stress fiber formation, demonstrating a causal relationship (Supplementary Fig. 10).
2. In ROCK knock-in cells, endogenous GFP-tagged ROCK localized from the plasma membrane to arc stress fibers (Fig. 5e-i), and higher ROCK intensity at the frontal protrusion correlated with spontaneous directional changes, indicating a structural

mechanism for polarity breaking and orthogonal turning.

3. The functional importance of arc stress fibers in regulating migration was validated across multiple cell lines (Supplementary Fig. 15), supporting the general role of Rac-ROCK-driven arc stress fiber formation in motility regulation.

Taken together, these results provide robust and physiologically relevant evidence supporting the critical role of Rac-ROCK interactions in governing cell migration behavior.

Additionally, the experiments using ROCK inhibitors lack specificity. The global inhibition of all ROCK molecules is not to test the authors claim of ROCK-Rac interactions in a specific subcellular location. A more precise approach would involve selectively disrupting the interaction between ROCK and Rac, and similarly for other interactions such as CDC42 and FMNL, to provide more definitive evidence.

We thank the reviewer for raising the important point regarding the specificity of ROCK inhibition experiments. To directly address this, we employed the Rac1F37W mutant in this study, which was specifically designed to selectively disrupt the interaction between Rac1 and ROCK. The F37W substitution alters the effector-binding interface of Rac1, based on structural insights from the RhoA-ROCK complex (PDB: 1S1C, Supplementary Fig. 13a), selectively impairing ROCK binding while preserving interactions with other effectors (Fig. 4a). Therefore, we consider that Rac1F37W is a precise tool for perturbing the Rac-ROCK interaction, directly responding to the reviewer's suggestion. Consistent with this, both overexpression of constitutively active Rac1(Q61L) and chemically inducible activation of endogenous Rac1 in ROCK1/2 double knockout cells failed to induce arc stress fiber formation (Supplementary Fig. 5e and f), further supporting the critical role of Rac-ROCK interaction in arc stress fiber assembly and related migratory behavior.

Regarding the Cdc42-FMNL interaction, we included in the revised manuscript experiments examining previously reported switch-of-function mutants (PMID: 12732140), in which swapping five amino acids between Rac1 and Cdc42 alters their effector binding properties. These mutants displayed distinct changes in binding specificity: the Rac1 5 a.a. mutant lost ROCK interaction but gained FMNL binding, whereas the Cdc42 5 a.a. mutant showed markedly reduced FMNL interaction. These alterations were accompanied by reversed cellular morphologies—Rac1 5 a.a. mutant cells exhibited diminished lamellipodia and increased filopodia, likely due to the gain of FMNL and loss of ROCK interactions (Fig. R1). These results underscore the functional specificity of Rac-ROCK and Cdc42-FMNL interactions in orchestrating actin cytoskeletal architecture.

Figure R1. ROCK and FMNL as key effectors transforming Rac1 and Cdc42 activities into distinct cellular morphology.

We have incorporated these data and corresponding discussion into the revised manuscript (Supplementary Fig. 5j and k; lines 160–170).

2. Clarity and specificity of the biological question

The manuscript states that “the intrinsic molecular machinery for spontaneous migration remains elusive,” but this question is too broad and loosely defined. Moreover, the presented data do not directly address this claim, at least not in an explicit manner. It is well-established in the field that cells generally utilize the same molecular machinery for migration regardless of external cues. If the authors propose that distinct machinery exists for cue-dependent and spontaneous migration, they should provide evidence to support this distinction.

As it stands, the study primarily focuses on Rho GTPases, indicating that the authors had a specific molecular target in mind from the outset. In that case, the central question would be better framed around the regulation of these GTPases, allowing for a clearer alignment between hypothesis and findings. Currently, there is a disconnect between the stated question and the experimental approach.

We thank the reviewer for the insightful comment. It is well established that Rho GTPases such as Cdc42 and Rac1 play key roles in cell migration. However, how these GTPases function and coordinate specifically during spontaneous migration in the absence of external directional cues remains unclear. In our study, we refined the central question to focus on the intrinsic regulation of spontaneous migration by Rho GTPases and their effectors. Using our INSPECT platform, we systematically analyzed these interactions to reveal how distinct coordination of Cdc42 and Rac1 contributes to directional persistence and spontaneous turning.

Accordingly, we have revised the abstract and main text to better reflect this focused biological question and the mechanistic insights our data provide (lines 21–25, 46–52).

3. Justification and novelty of the technological approach

Without a clearly articulated biological question, the justification for developing a new protein–protein interaction (PPI) detection tool is weak. Given that membrane lipids and other factors such as metabolites also contribute to the regulation of cell migration, it is not evident why PPI detection is the most appropriate or necessary method for addressing the biological question.

We appreciate the reviewer’s comment regarding the justification for developing our PPI detection tool. As addressed in our response to comment 2, we clarified the biological question framing, which motivated our approach. Our INSPECT platform enables simultaneous, high-throughput comparison of multiple GTPase-effector interactions under identical cellular conditions. This parallel screening allowed us to detect functional differences between closely related GTPases, such as Rac1 and Cdc42, with high sensitivity and specificity. While we recognize that other factors, including membrane lipids and metabolites, also regulate cell migration, our study focuses specifically on dissecting the intricate role of the Rho GTPase-effector ensemble in orchestrating intrinsic motility programs.

Thus, INSPECT provides an opportunity to quantitatively analyze these direct protein-protein interactions in living cells, enabling insights that cannot be easily obtained by alternative approaches.

Furthermore, the INSPECT technology appears to be a rebranding of previous work by the same group (e.g., PMID: 20133723 and 21796746). The manuscript does not clearly explain how INSPECT differs from or improves upon these earlier methods. If the modifications were essential for identifying novel Rho partners, that rationale should be explicitly stated and

experimentally supported. Without this clarification, the novelty of the method remains questionable.

We appreciate the reviewer's constructive feedback and the opportunity to clarify the novelty of the INSPECT platform compared to our previous methods (e.g., PMID: 20133723, 21796746).

INSPECT introduces three key improvements over previously developed methods:

1. Distinguishing true bait-prey interactions from homo-interactions
: INSPECT enhances specificity by conjugating bait and prey proteins to distinct multimeric scaffolds. While condensates can form from bait-bait or prey-prey homo-interactions, INSPECT allows clear discrimination between these and true bait-prey interactions. Because condensates driven by homo-interaction involve only one scaffold type, whereas true bait-prey interactions bring together both scaffold types into mixed condensates, the two can be easily distinguished based on scaffold composition. This design minimizes false-positive interpretations and ensures that observed condensate formation reflects genuine Rho GTPase-effector interactions.
2. Chemical-free induction of phase separation
: In contrast to earlier methods relying on rapamycin-induced dimerization, INSPECT detects protein-protein interactions without any chemical inducers. This design eliminates potential artifacts arising from perturbation of endogenous signaling pathways such as mTOR, allowing PPIs to be assessed under more physiological conditions.
3. Simplified and more robust system
: INSPECT requires expression of only two components instead of three, as in previous systems. This simplification reduces variability caused by differential expression of multiple constructs, improves experimental robustness, and enhances ease of use.

These improvements were critical for reliably identifying Rho-effector interactions in this study. We have revised the manuscript to explicitly highlight these technical advances (lines 397–409).